# A Systematic Study of In-the-Wild Model Merging for Large Language Models

**Oğuz Kağan Hitit**\*                                          *ohitit20@ku.edu.tr*
*Koç University*

**Leander Girrbach**                                          *leander.girrbach@helmholtz-munich.de*
*Technical University of Munich*
*Munich Center for Machine Learning*
*Helmholtz Munich*

**Zeynep Akata**                                          *zeynep.akata@helmholtz-munich.de*
*Technical University of Munich*
*Munich Center for Machine Learning*
*Helmholtz Munich*

**Reviewed on OpenReview:** *https://openreview.net/forum?id=6zSIyrqS7J*

## Abstract

Model merging combines multiple fine-tuned checkpoints into a single model without additional training, offering an attractive approach to reusing models and efficiently improving performance. However, it remains unclear whether the advantages reported for settings where all merged experts have distinct roles and are tuned on clearly separated tasks also hold in settings where the merged experts do not have clearly distinct roles, but are trained on overlapping or even conflicting objectives. To evaluate this setting, we present a large-scale, systematic evaluation of "in-the-wild" model merging of heterogeneous experts, that may have been trained on overlapping or conflicting objectives. Concretely, we evaluate six state-of-the-art merging methods, including recent subspace methods, across four open-weight LLMs, twelve fine-tuned checkpoints per base model, and sixteen standard LLM benchmarks. Evaluating through standardized benchmarks, we measure both the probability that a model merged from a heterogeneous set of experts outperforms the base model and we measure relative gains over the best individual checkpoint. Our results show that the oldest and simplest method, Task Arithmetic, is the only approach that reliably yields performance gains on LLMs in this "in-the-wild" setting. Other interference-aware and subspace merging methods typically do not result in notable improvements over the base model. Our findings indicate that current merging techniques mostly do not enable extracting useful weight updates from heterogeneous and potentially conflicting versions. This motivates the design of LLM-specific merging algorithms and merging-aware fine-tuning methods. Code is available at `https://github.com/kaganhitit11/mergeval`.

## 1 Introduction

Recently, model merging has gained considerable attention due to its empirically strong efficacy in combining different models with the same architecture. Among the most intriguing observations is the phenomenon of *constructive interference*, where a merged model outperforms its individual base models (Stojanovski et al., 2022; Yadav et al., 2023; Roth et al., 2024). In this paper, we focus on a specific instantiation of this phenomenon: whether we can improve the general capabilities of large language models (LLMs) by merging a heterogeneous set of "in-the-wild" fine-tuned versions. Answering this question is interesting because, if

---

\*Work done while at Technical University of Munich.

possible, it allows us to derive an improved version of the base model with minimal additional cost. This model can, in turn, be used to produce stronger specialized versions, like the setup in (Roth et al., 2024). Previous work (He et al., 2025; Cohere et al., 2025) has shown that final merged models can retain significant in-domain performance of the individual merged checkpoints but do not surpass them. Orthogonal to this setting, our research evaluates whether merging multiple heterogeneous, potentially overlapping or conflicting, checkpoints can produce models that outperform any individual checkpoint, as measured by average performance across a wide range of tasks. In summary, we evaluate whether merging heterogeneous models can lead to an overall improved model, beyond equipping the base model with task-specific performance from one or more fine-tuned versions (He et al., 2025).

Understanding this question is important for both scientific and practical reasons. On the practical side, organizations often accumulate dozens of fine-tuned checkpoints tailored to specific domains, tasks, or use cases. These checkpoints do not necessarily harmonize or contribute towards a common improvement direction. If they can jointly improve the underlying model beyond any individual checkpoint, this provides a practical application for their reuse. Additionally, understanding which methods and settings enable this kind of improvement provides insight into how knowledge is distributed in the parameter space of LLMs, offering clues about the geometry of fine-tuning and the limitations of weight-space interpolation.

In this study, we address this gap by conducting a large-scale, systematic evaluation of state-of-the-art model merging techniques across multiple LLM families, a heterogeneous, "in-the-wild" set of fine-tuned checkpoints, and a wide suite of benchmarks. Our work seeks to answer the following research questions: (1) Can we produce an improved version of the base LLM by simply merging multiple fine-tuned versions? (2) Which weight interpolation-based model merging techniques enable such improvement? (3) Do recently proposed merging methods that operate on the subspaces of weight matrices also improve performance of "in-the-wild" merging in LLMs?

In summary, our main contributions are: (1) We systematically evaluate six model merging methods on four LLMs across 16 benchmarks; (2) We find that most merging methods do not produce models that outperform all involved individual checkpoints. This motivates further research on how to leverage capabilities of existing heterogeneous model versions and how to combine them; (3) Among all six evaluated merging methods, only *Task Arithmetic*, the oldest and simplest of the methods, consistently yields models that outperform all involved individual checkpoints. However, performance gains are limited. The partial success of Task Arithmetic shows that even in heterogeneous pools of model versions, there is complementary knowledge that can be used to improve the base model, but it is non-trivial to extract, and more sophisticated merging methods are not better suited to do so.

These claims are supported by extensive experiments: We evaluate four LLMs, spanning different model families (Qwen3 and Llama3) and different model sizes (3B, 4B, and 8B), on 16 standard LLM benchmarks, which allows for generalizable insights. Observed trends are consistent across the evaluated models and benchmarks, so they can be assumed to hold for other models as well. Finally, our insights are relevant to the model merging and broader machine learning community, as a systematic evaluation of subspace merging methods on LLMs has been lacking so far, and our results are likely to inspire future research on model merging, specifically targeting LLMs and heterogeneous, "in-the-wild" merging.

## 2   Related Work

Model merging for LLMs has been surveyed extensively. Li et al. (2025b) review model fusion across architectures and disjoint training runs. Yang et al. (2026) group LLM merging approaches into "Pre-Merging Methods" (weight alignment), "During-Merging Methods" (weight combination), and "Theories and Analysis". We use "merging methods" to denote the second category. Ruan et al. (2025) classify merging approaches with emphasis on pruning, while Yadav et al. (2025) systematically study merging across model scales up to 64B parameters. Our work complements these analyses by incorporating additional recent subspace methods and evaluating widely used open-weight models (Qwen3 and Llama 3) rather than proprietary PaLM models.

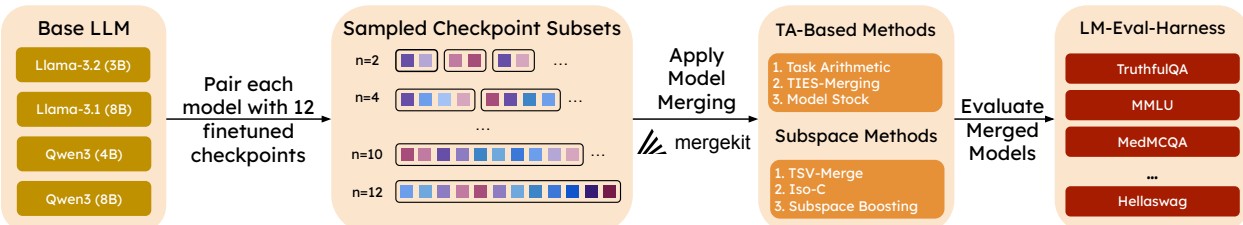

Figure 1: Our evaluation protocol pairs each base large language model (LLM) with 12 publicly available heterogeneous, i.e. "in-the-wild", checkpoints and repeatedly samples subsets to merge. The sampled checkpoints are merged using three task arithmetic (TA) and three subspace merging methods. Resulting merged models are evaluated on 16 standard LLM benchmarks from lm-eval-harness to analyze trends in which merging methods consistently work well on LLMs.

## 2.1 Background on Motivations and Theoretical Foundations of Model Merging

Stochastic weight averaging (SWA) shows that combining weights from multiple checkpoints of the same model improves performance (Izmailov et al., 2018; Guo et al., 2023). By averaging points along a training trajectory, SWA benefits from mode connectivity (Draxler et al., 2018; Garipov et al., 2018; Kuditipudi et al., 2019; Benton et al., 2021), i.e. the observation that distinct optima are linked by low-loss paths. Thus, model variants sharing an optimization trajectory can be interpolated with negligible performance loss (Frankle et al., 2020). Robustness to small weight perturbations further supports such combinations (Arora et al., 2018). However, merging models trained from different bases requires neuron alignment (Tatro et al., 2020; Entezari et al., 2022), and several methods address this (Ainsworth et al., 2023; Peña et al., 2023; Rinaldi et al., 2025). Here, however, we restrict our focus to fine-tuned LLM checkpoints derived from a common base and therefore do not consider neuron alignment.

## 2.2 Detailed Overview of Model Merging Techniques and Paradigms

**Weight Interpolation Based Methods.** Wortsman et al. (2022) introduce *Model Soup*, which averages or greedily aggregates aligned models. For fine-tuned variants of a shared base, Ilharco et al. (2023) propose *Task Arithmetic (TA)*, a main method in our study (detailed in Section 3.1). Several approaches refine TA to reduce interference across merged models. *DARE* (Yu et al., 2024) drops a fraction of delta parameters and rescales the rest, and *DAREx* (Deng et al., 2025) adapts this for extreme pruning rates. *DELLA* (Deep et al., 2024) prunes by magnitude, preserves consistent parameter signs, and fuses selected updates. *Model Breadcrumbs* (Davari & Belilovsky, 2024) applies layer-wise masking to remove large outliers and small noise, while *EMR-Merging* (Huang et al., 2024b) masks and rescales task vectors individually. *TIES-Merging* (Yadav et al., 2023) trims small updates, enforces sign consensus, and merges only aligned parameters. SLERP (Shoemake, 1985) performs geodesic interpolation to preserve geometric structure.

**Training-Based Methods.** Others optimize parameters such as interpolation coefficients, for instance *LoraHub* (Huang et al., 2024a) merges LoRA adapters (Hu et al., 2022) via weighted averaging with gradient-free coefficient tuning on validation data. Routing-based methods combine components in MoE architectures (Kang et al., 2025; Li et al., 2024a; Muqeeth et al., 2024; Tang et al., 2024a; Lu et al., 2024). Additional techniques use data statistics or validation sets to select averaging coefficients (Yang et al., 2024b; Zhou et al., 2024; Zhang et al., 2024; Li et al., 2025a), pruning masks (Wang et al., 2024; Tang et al., 2023; Kong et al., 2024), or parameter rescaling (Matena & Raffel, 2022; Jin et al., 2023; Daheim et al., 2024). Akiba et al. (2025) optimize merging strategies via evolutionary search. Post-training or model linearization can further improve mergeability (Yang et al., 2024a; Ortiz-Jimenez et al., 2023; Tang et al., 2024b; Liu et al., 2024).

**Subspace Merging Methods.** Recent approaches treat merging as a problem within low-rank task subspaces rather than full parameter space. Skorobogat et al. (2025) address the rank collapse of task vectors

with *subspace-boosted merging*, using SVD to preserve expressive directions. In parameter-efficient fine-tuning (PEFT), Stoica et al. (2025) introduce *KnOTS*, which aligns LoRA-based updates into a shared subspace to improve compatibility. Marczak et al. (2025) analyze singular value spectra to decompose updates into common and task-specific subspaces, mitigating interference. Tam et al. (2024) frame merging as solving linear systems in task parameter subspaces. Finally, Gargiulo et al. (2025) use per-layer SVD to isolate task-relevant directions, showing that singular vectors can guide merging to reduce destructive interference.

**Constructive Interference.** *Constructive interference* is the main focus of this study. It occurs when a merged model outperforms its constituent experts by leveraging their complementary strengths. Wortsman et al. (2022) show that averaging fine-tuned weights improves generalization compared to single checkpoints. Ilharco et al. (2023) demonstrate that linear combinations of task vectors enable transfer and domain generalization. Yadav et al. (2023) highlight that resolving weight conflicts produces merged models that consistently outperform their parents. Similar findings exist in reinforcement learning (Ramé et al., 2023b; 2024b) and continual learning (Stojanovski et al., 2022). However, most evaluations focus on moderate-scale Transformers like BERT (Devlin et al., 2019) or T5 (Raffel et al., 2020), leaving the generalization to modern large-scale LLMs an open question.

## 2.3 Practical Applications of Model Merging

Model merging naturally enables multi-task models derived from task-specific variants (Wang et al., 2024; Matena & Raffel, 2022; Daheim et al., 2024). For example, Awasthy et al. (2025) build a strong teacher for distillation by merging models trained on different objectives. Merging also mitigates catastrophic forgetting during fine-tuning and continual learning, helping models retain base-model knowledge (Alexandrov et al., 2024; Porrello et al., 2025; Zhu et al., 2024; Marczak et al., 2024; Xiao et al., 2024; Chitale et al., 2023; Qazi et al., 2024; Stojanovski et al., 2022). Weight averaging further enhances out-of-distribution (Izmailov et al., 2018; Ramé et al., 2022; 2023a; 2024b; Jolicoeur-Martineau et al., 2024; Jain et al., 2023; Li et al., 2025c) and out-of-domain generalization (Arpit et al., 2022; Li et al., 2024b), strengthening robustness to adversarial attacks and jailbreaks (Cong et al., 2023; Croce et al., 2023; Gallego, 2024). Finally, merging supports instruction tuning and alignment of RLHF-tuned LLMs (Fu et al., 2024; Ramé et al., 2024a).

# 3 Do Methods Based on Task Arithmetic Enable Constructive Interference?

Our goal is to systematically evaluate if existing model merging techniques can achieve constructive interference in LLMs by merging heterogeneous fine-tuned versions. We focus on methods similar to the seminal Task Arithmetic method (Ilharco et al., 2023), which merge models by interpolating their weights. Our evaluation includes three merging techniques, four base LLMs, 12 fine-tuned versions of each LLM, and 16 benchmark tasks. This allows us to provide a comprehensive overview of the strengths and limitations of merging methods when applied to in-the-wild merging of LLMs.

## 3.1 Merging Methods in this Study: Task Arithmetic, TIES-Merging, and Model Stock

We evaluate three popular algorithms that represent distinct paradigms for model merging: Task Arithmetic (Ilharco et al., 2023), TIES-Merging (Yadav et al., 2023), and Model Stock (Jang et al., 2024). These methods respectively capture linear vector arithmetic, interference-aware adjustment, and geometric interpolation. We do not include other recent approaches such as Consensus Merging (Wang et al., 2024) or Model Soups (Wortsman et al., 2022), as these methods are likely to perform similarly to simple averaging under large-scale conditions or rely on domain-specific heuristics that make systematic comparison difficult. In the following, we briefly introduce all evaluated merging methods, and we visualize them in Fig. 2.

**Task Arithmetic.** Task Arithmetic (Ilharco et al., 2023) frames model merging as vector addition and subtraction in weight space, treating fine-tuning updates as *task vectors*. Given a base model $W_0$ and its fine-tuned variant $W_i$, the corresponding task vector is defined as

$$\Delta W_i = W_i - W_0. \tag{1}$$

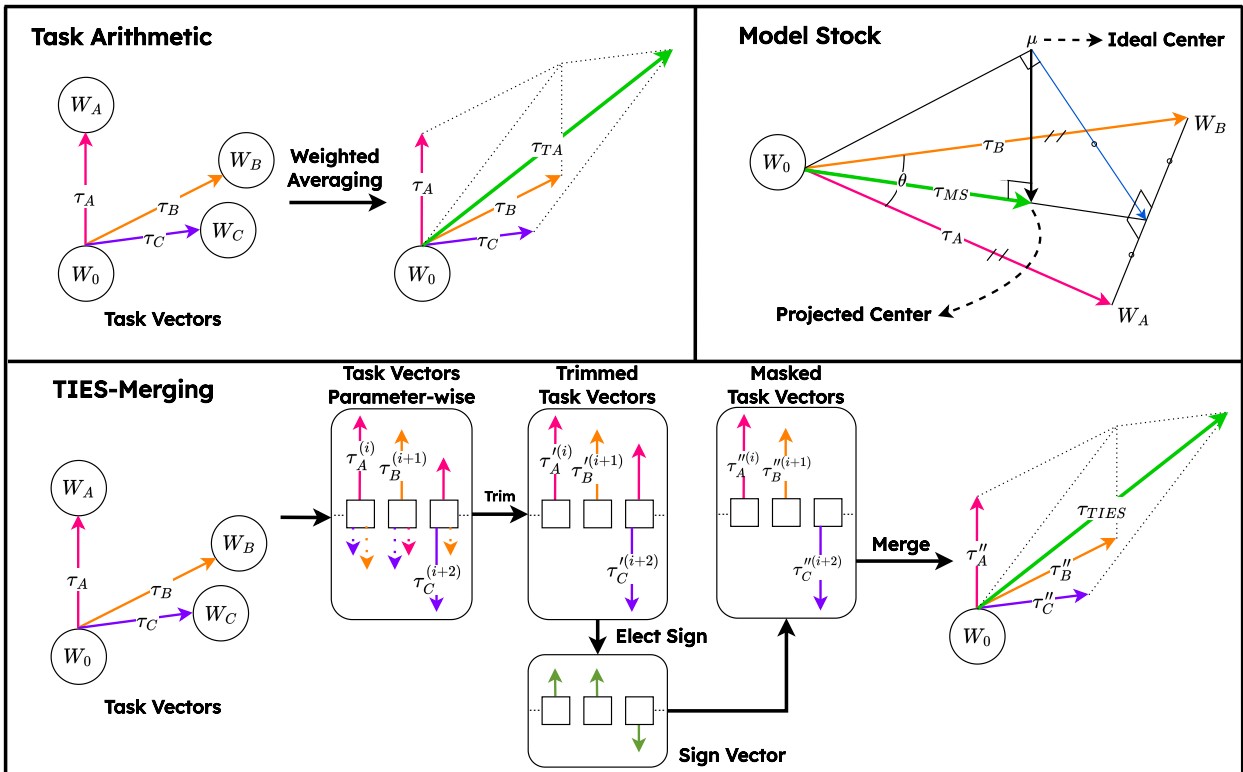

Figure 2: Overview of task-arithmetic–based model merging methods: Task Arithmetic, TIES-Merging, and Model Stock. Given a base model $W_0$ and fine-tuned checkpoints $W_i$, *Task Arithmetic* computes task vectors $\Delta W_i = W_i - W_0$ and merges them via weighted addition. *TIES-Merging* extends this by (1) trimming small-magnitude parameter updates, (2) enforcing sign-consistent updates across checkpoints, and (3) merging only aligned parameters to reduce interference. *Model Stock* instead interpolates between $W_0$ and the geometric center of the fine-tuned checkpoints based on estimated inter-model angles.

These task vectors encode learned task-specific knowledge and can be algebraically combined to transfer, compose, or remove capabilities across models. A merged model $W_{\text{merged}}$ can thus be expressed as

$$\Delta W_{\text{TA}} = \sum_{i=1}^{n} \alpha_i \Delta W_i, \qquad W_{\text{merged}} = W_0 + \lambda \Delta W_{\text{TA}}, \tag{2}$$

where $\alpha_i$ denotes the coefficient assigned to each expert model, and $\lambda$ is a global, scalar scaling factor. Setting $\alpha_i = 1$ for a target task and $\alpha_j = -1$ for an undesired task allows additive or subtractive transfer, respectively, enabling "forgetting by negation" and "learning by addition". In our experiments, we set $\alpha_i = 1$, and $\lambda = 1$ for all checkpoints.

**TIES-Merging.** TIES (Yadav et al., 2023) also uses task vectors, but attempts to mitigate conflicts between merges in weight space. Given a set of fine-tuned weights $\{W_i\}_{i=1}^{n}$ and a common initialization $W_0$, each task vector $\Delta W_i$ is defined as in Task Arithmetic (Eq. (1)). The method proceeds in three stages. *(1) Trim:* within each layer, only the top-$k\%$ of parameters in $\Delta W_i$ based on absolute magnitude are retained, and the rest are reset to zero, producing a sparsified update $\Delta W_i^{\text{trimmed}}$. This step removes weak or noisy signals. *(2) Select signs:* for each parameter, a sign consensus across all checkpoints $\Delta W_i^{\text{trimmed}}$ is computed. Parameters in $\Delta W_i^{\text{trimmed}}$ whose sign disagrees with the sign consensus are masked out, yielding $\Delta W_i^{\text{masked}}$. This sign selection ensures that only updates with consistent directional agreement contribute to the merge, while conflicting parameters are reset to the base value. *(3) Disjoint merge:* Similar to Task Arithmetic, the

final merged model is computed as

$$\Delta W_{\mathrm{TIES}} = \frac{1}{n} \sum_{i=1}^{n} \alpha_i \Delta W_i^{\mathrm{masked}}, \qquad W_{\mathrm{merged}} = W_0 + \lambda \Delta W_{\mathrm{TIES}}. \tag{3}$$

Intuitively, TIES preserves the relevant task updates while filtering out contradictory ones.

**Model Stock.** Model Stock (Jang et al., 2024) moves the merged weights toward the geometric center of a set of fine-tuned checkpoints: given pre-trained weights $W_0$ and fine-tuned checkpoints $\{W_i\}_{i=1}^{N}$, Model Stock selects the point that is geometrically closest to the unknown center, i.e. the true geometric midpoint of the shell that the checkpoints would define in weight space by a layerwise interpolation between $W_0$ and the average of the fine-tuned variants ($W_{\mathrm{avg}}$). Mathematically, the method computes the merged model as

$$W_{\mathrm{avg}} = \frac{1}{N} \sum_{i=1}^{N} W_i, \qquad t = \frac{N \cos \theta}{1 + (N-1)\cos\theta}, \qquad W_{\mathrm{merged}} = t \, W_{\mathrm{avg}} + (1-t) \, W_0. \tag{4}$$

where $N$ denotes the number of fine-tuned variants, $t$ denotes the interpolation factor, $\theta$ denotes the mean inter-model angle (measured layerwise) among the fine-tuned variants. When the checkpoints are tightly aligned (small $\theta$), $t$ is larger and the merge relies more on $W_{\mathrm{avg}}$; when they are more diverse (large $\theta$), $t$ decreases and the merge leans toward $W_0$. We acknowledge that Model Stock, in its original formulation, is intended to merge multiple checkpoints from the *same* training trajectory. However, formally, there is no constraint against applying Model Stock to checkpoints fine-tuned on different datasets. Thus, we include it in our comparison for a more complete comparison, and as an explicit "stress-test" for Model Stock.

## 3.2 Experimental Setup

**Models and Checkpoints.** We evaluate four open-weight LLMs spanning two families and parameter scales: LLAMA 3.2 3B, LLAMA 3.1 8B (Dubey et al., 2024), QWEN3 4B, and QWEN3 8B (Yang et al., 2025). This diversity supports generalizable conclusions. For each base model, we merge 12 publicly available fine-tuned checkpoints that cover various objectives and domains (Appendix A). Merging methods use *mergekit* (Goddard et al., 2024) with hyperparameters fixed to values identified in Appendix C. We set $\lambda = 1.0$ for Task Arithmetic and Model Stock, and $\lambda = 0.1$ for TIES-Merging. We use top-10% magnitude threshold for TIES-Merging.

**Sampling Checkpoints to Merge.** To study how performance scales with the number of merged models, we follow a progressive merging strategy. For each base model and method, we evaluate the base model, all 12 individual fine-tuned checkpoints, and merged models containing (2, 4, 6, 8, 10) and (12) checkpoints. Because the number of possible combinations grows combinatorially, we uniformly sample 15 subsets for each merge size and report the mean performance. The same subsets are used across methods, ensuring differences arise from the merging algorithms rather than checkpoint selection.

## 3.3 Evaluation on Standard LLM Benchmarks

**Benchmarks.** We evaluate every base model and merged configuration with the *lm-evaluation-harness* library (Biderman et al., 2024), using its standardized implementations for the following Open LLM Leaderboard tasks: `arc_easy`, `arc_challenge`, `hellaswag`, `winogrande`, `boolq`, `piqa`, `openbookqa`, `commonsense_qa`, `headqa`, `prost`, `truthfulqa_mc1`, `mmlu`, `medmcqa`, `leaderboard_gpqa`, `leaderboard_bbh`, and `leaderboard_mmlu_pro`. These benchmarks collectively cover multiple evaluation axes including commonsense and scientific question answering (e.g., `commonsense_qa`, `medmcqa`), multi-step reasoning (e.g., `arc_challenge`, `bbh`), and instruction-following (e.g., `hellaswag`, `winogrande`). We use the default decoding setup and per-task n-fewshot configuration of *lm-eval-harness* for all benchmarks. We do not apply any chat templates, and we report accuracy for each task. The exact n-fewshot values are provided in Appendix F.

**Results.** In Fig. 3, we show the average performance across benchmarks for all merged models and merging methods (task-wise accuracies are in Appendix B). For merging methods, we notice clear trends that hold

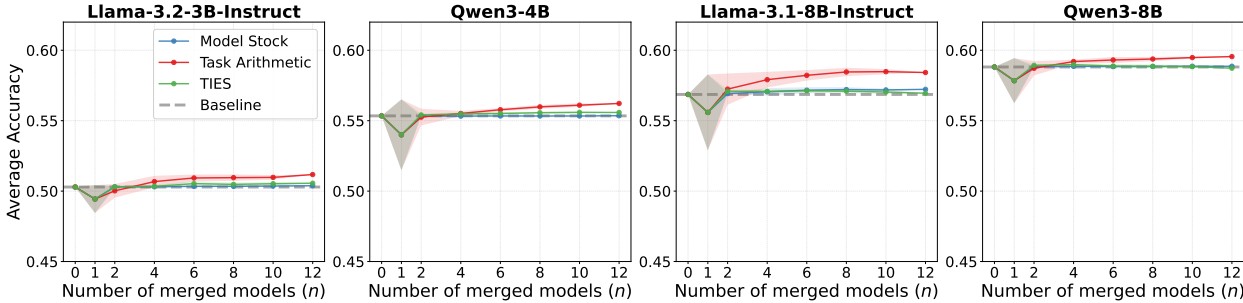

Figure 3: Average accuracy and standard deviation of the models across all benchmarks. From left to right, models are LLAMA 3.2 3B, QWEN3 4B, LLAMA 3.1 8B, QWEN3 8B, respectively. Shaded areas indicate the standard deviation over different samples of merged checkpoints.

regardless of the model. *Task Arithmetic* steadily improves as more experts are combined, becoming reliably superior to the base model once a moderate number of experts are merged. This clearly demonstrates the existence of constructive interference in LLMs: merging several independent fine-tuned checkpoints can produce a model that surpasses both the base LLM and any individual expert. At the same time, the improvement achieved through merging is modest, generally less than 1% averaged over all tasks. At most, we can achieve 13.07% improvement for `prost` task in LLAMA 3B when all twelve checkpoints are merged with Iso-C. *Model Stock* does not deviate significantly from the performance of the base model, and also weights stay very close to the base model. This shows its limited ability in finding interpolations of different fine-tuned versions that generalize better. Finally, *TIES-Merging*, despite building on top of Task Arithmetic and using a more sophisticated approach, remains tightly clustered around the base model's performance. While it demonstrates improvements similar to Model Stock, it consistently falls short of the gains achieved by Task Arithmetic, maintaining a lower average accuracy across all merged model counts.

These observations are quantified in Table 1, which reports both the probability of surpassing the base model and the corresponding relative improvement for each $n$. Across all four models, *Task Arithmetic* exhibits a clear, monotonic trend: both the success probability and the relative gain steadily increase as more models are merged. For example, for LLAMA 3B, TA improves over the base model in only 20% of combinations at $n=2$, but already reaches 80% at $n=4$ and 100% for all $n \geq 6$, with the average relative improvement rising from $-0.27$ at $n=2$ to $+0.89$ at $n=12$. This pattern consistently appears in the other models as well: TA reaches 100% success for all $n \geq 4$ in LLAMA 8B and all $n \geq 6$ in both Qwen models, with relative gains reaching as high as $+1.62$ (LLAMA 8B, $n=10$) and $+0.88$ (QWEN 4B, $n=12$). *Model Stock* follows a similar but weaker pattern: improvements are small but consistently positive at higher $n$ values, aligned with its conservative update rule. For instance, LLAMA 8B shows gains growing from $+0.06$ at $n=2$ to $+0.36$ at $n=12$, and $+0.36$ is the highest relative improvement that Model Stock achieves across all models. *TIES-Merging* achieves a high probability of improving over the base model (averaging 75–87% across $n \geq 2$), but the magnitude of these gains remains small and plateaus quickly, with the average relative improvement hovering around $+0.17$. TIES also exhibits instability at higher merge counts for certain models. For QWEN-8B, performance degrades from a peak of $+0.16$ at $n = 4$ to $-0.08$ at $n = 12$. As discussed in Appendix C, this behavior is likely a consequence of the method's sensitivity to task vector magnitude in heterogeneous settings.

It is also important to note that individual fine-tuned checkpoints rarely outperform their own base model: at $n=1$, fewer than 50% of the checkpoints exceed the accuracy of their corresponding base. In other words, a randomly selected expert is more likely to underperform than improve upon the base model. This confirms that the gains observed at higher $n$ do not stem from simply picking stronger experts, but rather from the constructive interference produced by merging multiple weaker ones.

Beyond improvements over the base model, we also examine whether merging can surpass the strongest individual fine-tuned checkpoint. As shown in Table 2, *Task Arithmetic* reliably exceeds the best expert for

| Model | Method | Base | $n$=1 (12) | $n$=2 (15) | $n$=4 (15) | $n$=6 (15) | $n$=8 (15) | $n$=10 (15) | $n$=12 (1) |
|---|---|---|---|---|---|---|---|---|---|
| | TA | 50.3 | 17 / -0.85 | 20 / -0.27 | 80 / +0.39 | 100 / +0.64 | 100 / +0.66 | 100 / +0.68 | 100 / +0.89 |
| LLAMA-3B | Model Stock | 50.3 | 17 / -0.85 | 60 / +0.01 | 87 / +0.02 | 87 / +0.05 | 100 / +0.05 | 93 / +0.07 | 100 / +0.09 |
| | TIES | 50.3 | 17 / -0.85 | 47 / +0.02 | 60 / +0.04 | 87 / +0.24 | 87 / +0.18 | 87 / +0.23 | 100 / +0.27 |
| | TA | 56.9 | 25 / -1.28 | 60 / +0.38 | 93 / +1.05 | 100 / +1.36 | 100 / +1.60 | 100 / +1.62 | 100 / +1.56 |
| LLAMA-8B | Model Stock | 56.9 | 25 / -1.28 | 93 / +0.06 | 100 / +0.19 | 100 / +0.30 | 100 / +0.35 | 100 / +0.32 | 100 / +0.36 |
| | TIES | 56.9 | 25 / -1.28 | 100 / +0.23 | 93 / +0.21 | 100 / +0.25 | 100 / +0.23 | 93 / +0.17 | 100 / +0.08 |
| | TA | 55.3 | 25 / -1.34 | 47 / -0.09 | 80 / +0.17 | 100 / +0.44 | 100 / +0.64 | 100 / +0.76 | 100 / +0.88 |
| QWEN-4B | Model Stock | 55.3 | 25 / -1.34 | 13 / -0.01 | 20 / -0.02 | 47 / -0.01 | 27 / -0.01 | 27 / -0.01 | 100 / +0.01 |
| | TIES | 55.3 | 25 / -1.34 | 66 / +0.06 | 93 / +0.14 | 100 / +0.17 | 100 / +0.22 | 100 / +0.25 | 100 / +0.24 |
| | TA | 58.8 | 33 / -0.97 | 67 / -0.09 | 100 / +0.39 | 100 / +0.50 | 100 / +0.56 | 100 / +0.67 | 100 / +0.74 |
| QWEN-8B | Model Stock | 58.8 | 33 / -0.97 | 47 / +0.01 | 100 / +0.04 | 93 / +0.03 | 100 / +0.03 | 100 / +0.05 | 100 / +0.05 |
| | TIES | 58.8 | 33 / -0.97 | 100 / +0.11 | 93 / +0.16 | 60 / +0.06 | 53 / +0.04 | 40 / +0.02 | 0 / -0.08 |
| | TA | 55.3 | 25 / -1.11 | 49 / -0.02 | 88 / +0.50 | 100 / +0.74 | 100 / +0.87 | 100 / +0.93 | 100 / +1.02 |
| **Average** | Model Stock | 55.3 | 25 / -1.11 | 53 / +0.02 | 77 / +0.06 | 82 / +0.09 | 82 / +0.10 | 80 / +0.11 | 100 / +0.13 |
| | TIES | 55.3 | 25 / -1.11 | 78 / +0.11 | 85 / +0.14 | 87 / +0.18 | 85 / +0.17 | 80 / +0.17 | 75 / +0.13 |

Table 1: Constructive interference results for Task Arithmetic-based merging methods applied to models. Each entry contains two quantities: the percentage of merge combinations that exceed the base model's accuracy, and the mean relative accuracy improvement for those combinations. Column headers use the notation $n = m\,(k)$, where $n$ is the number of models merged and $k$ is the number of evaluated merge combinations for that value of $n$. Base indicates base model accuracy.

three of the four model families once $n \geq 4$. For example, in QWEN-4B, TA delivers a +1.02 improvement at $n$=4, which increases steadily to +1.72 at $n$=12. QWEN-8B shows an almost identical pattern, with gains rising from +1.14 at $n$=4 to +1.49 at $n$=12. LLAMA-3B also surpasses its best expert once $n \geq 4$, improving from +0.32 at $n$=4 to +0.82 at $n$=12. The only exception is LLAMA-8B, whose strongest fine-tuned checkpoint is unusually strong: merging never exceeds it, although the deficit shrinks meaningfully—from $-1.76$ at $n$=2 to only $-0.58$ at $n$=12. These results demonstrate that heterogeneous, "in-the-wild" model merging frequently produces models that outperform not only the base model but also the best available fine-tuned checkpoint in general capabilities.

To better understand the mechanism behind these performance differences, we measure the magnitude of the task vector, $\|\theta_{\mathrm{merged}} - \theta_{\mathrm{base}}\|_2$, as a function of $n$ in Fig. 4. Across all model families, Task Arithmetic, Task Arithmetic with Subspace Boosting, TIES, TIES with Subspace Boosting, and Model Stock remain very close to the base model, with task-vector norms generally below 50 for all $n$. In contrast, Iso-C and TSV-M produce substantially larger deviations, with distances often in the 100–300 range for LLAMA-3B and QWEN-4B, and exceeding 300 for LLAMA-8B and QWEN-8B. These displacements in parameter space correlates strongly with the performance degradation observed in Fig. 3 and Fig. 6, supporting the hypothesis that merging algorithms that aggressively change the weights and move outside the base model's loss basin are responsible for the observed catastrophic forgetting.

## 4 Do Subspace Merging Methods Enable Constructive Interference?

In Section 3, we found that only Task Arithmetic consistently achieves constructive interference in LLMs when merging heterogeneous experts, whereas Model Stock and TIES Merging, which are alternative methods operating in weight space, do not yield significant gains. However, recently, subspace-based model merging methods have achieved significant improvements when applied to vision-language models. Unlike weight interpolation methods that directly operate in full parameter space, subspace-based approaches merge models by aligning or projecting their task updates into subspaces. This approach mitigates rank collapse, isolates compatible update directions, and improves robustness during model composition. Therefore, we also evaluate subspace-based model merging methods, which have been primarily evaluated on vision-language

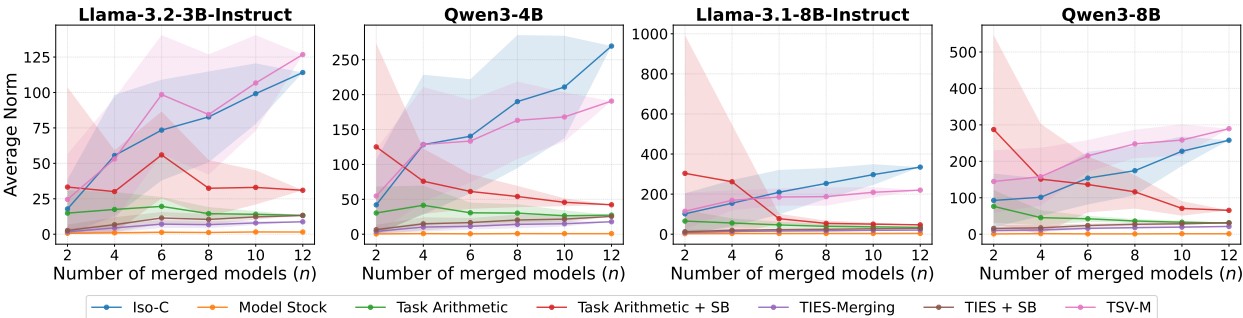

Figure 4: Average $L_2$-norm of the task vectors with respect to the base model as a function of the number of merged checkpoints. Each curve reports the mean Euclidean distance $\|\theta_{\mathrm{merged}} - \theta_{\mathrm{base}}\|_2$ across samples of merged models, with shaded regions indicating the standard deviation. Higher values indicate larger deviations from the base model in parameter space.

| Model | Best FT | $n$=1 (12) | $n$=2 (15) | $n$=4 (15) | $n$=6 (15) | $n$=8 (15) | $n$=10 (15) | $n$=12 (1) |
|---|---|---|---|---|---|---|---|---|
| LLAMA-3B | 50.4 | 17 / -0.92 | 20 / -0.34 | 80 / +0.32 | 100 / +0.58 | 100 / +0.60 | 100 / +0.62 | 100 / +0.82 |
| LLAMA-8B | 59.0 | 17 / -3.41 | 13 / -1.76 | 0 / -1.09 | 0 / -0.78 | 0 / -0.54 | 0 / -0.52 | 0 / -0.58 |
| QWEN-4B | 54.5 | 75 / -0.50 | 93 / +0.76 | 100 / +1.02 | 100 / +1.28 | 100 / +1.48 | 100 / +1.60 | 100 / +1.72 |
| QWEN-8B | 58.1 | 67 / -0.22 | 80 / +0.66 | 100 / +1.14 | 100 / +1.25 | 100 / +1.32 | 100 / +1.42 | 100 / +1.49 |
| **Average** | 55.5 | 44 / -1.22 | 52 / +0.08 | 70 / +0.60 | 75 / +0.84 | 75 / +0.97 | 75 / +1.03 | 75 / +1.13 |

Table 2: Constructive interference results for Task Arithmetic comparing merged models to the best fine-tuned checkpoint across all bases. For each base model, the best fine-tuned checkpoint is selected based on its average performance across all evaluated tasks and is used as a fixed reference for all merge comparisons. Each cell reports (i) the percentage of merge combinations that surpass this best fine-tuned model and (ii) the mean relative accuracy difference. Column headers use the notation $n = m\,(k)$, where $n$ is the number of merged models and $k$ is the number of evaluated merge combinations.

models or small language models, such as T5, for LLM model merging, using the same setup introduced in Section 3. Below, we give a brief overview of the evaluated methods.

## 4.1 Merging Methods in this Study: TSV-M, Iso-C, Subspace Boosting

We assess three representative subspace-oriented merging methods, namely, TSV-Merge (Gargiulo et al., 2025), Iso-C (Marczak et al., 2025), and Subspace Boosting (Skorobogat et al., 2025).

**TSV-Merge.** TSV-Merge (Gargiulo et al., 2025) compresses each task's update into dominant low-rank directions, orthogonalizes them across tasks, and recombines the resulting bases into an interference-minimized update. Similar to Task Arithmetic, for each fine-tuned variant $i \in \{1, \ldots, T\}$, task vectors $(\Delta W_i^{(\ell)})$ are created for each layer $\ell$. Then, TSV-Merge computes SVD of every layer-wise task vector,

$$\Delta W_i^{(\ell)} \;=\; U_i^{(\ell)} \Sigma_i^{(\ell)} V_i^{(\ell)^\top}, \tag{5}$$

where the singular vectors $U_i^{(\ell)}$ and $V_i^{(\ell)}$ are called *Task Singular Vectors* (TSVs) and the diagonal entries of $\Sigma_i^{(\ell)}$ quantify their importance. TSV-Merge then retains only the top $\frac{1}{T}$ fraction of singular components for each $(i, \ell)$ to control capacity and suppress noise, keeping the highest-energy directions. Then, the truncated TSVs are aggregated (suppressing $\ell$ for brevity) by concatenation,

$$U \leftarrow [\,U_1 \mid U_2 \mid \cdots \mid U_T\,], \qquad \Sigma \leftarrow \mathrm{block\text{-}diag}(\Sigma_1, \ldots, \Sigma_T), \qquad V \leftarrow [\,V_1 \mid V_2 \mid \cdots \mid V_T\,]. \tag{6}$$

Because different tasks may emphasize overlapping directions, TSV-Merge removes this redundancy via an orthogonal Procrustes projection. Computing SVDs of the concatenated matrices $U$ and $V$, the closest

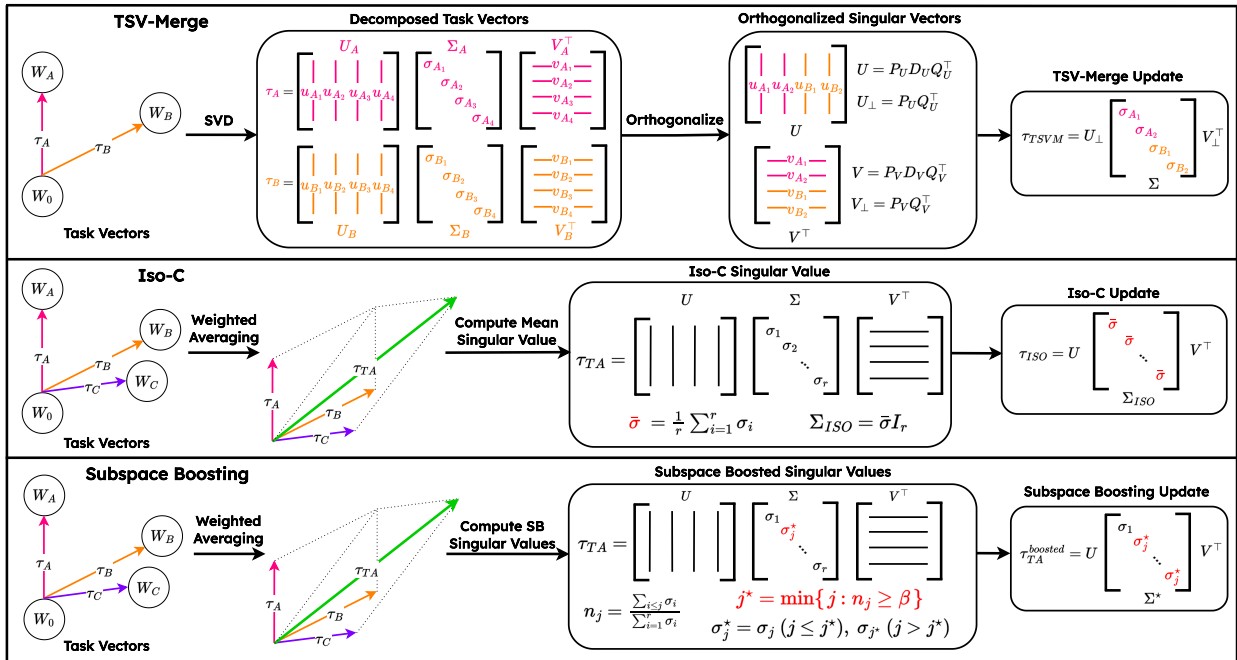

Figure 5: Overview of subspace-based model merging methods: TSV-Merge, Iso-C, and Subspace Boosting. These methods operate in low-rank task-update subspaces rather than full weight space. *TSV-Merge* extracts dominant singular directions for each task update, orthogonalizes them via Procrustes alignment, and recombines the aligned subspaces into a unified low-rank update. *Iso-C* flattens the singular value spectrum of the Task-Arithmetic update, producing an isotropically scaled representation of its principal directions. *Subspace Boosting* mitigates rank collapse by elevating weaker singular directions above a cumulative-energy threshold, broadening the effective subspace captured by the merged update. In the illustration, we show the TA+SB variant, but any task-vector-based merging method (e.g. TIES) could be substituted by modifying only how the merged task update is computed before applying the Subspace Boosting operation.

orthogonal factors in Frobenius norm are obtained in closed form as:

$$U = P_U D_U Q_U^\top, \qquad V = P_V D_V Q_V^\top, \qquad U_\perp = P_U Q_U^\top, \qquad V_\perp = P_V Q_V^\top. \tag{7}$$

With the aligned bases $U_\perp$ and $V_\perp$ in hand, TSV-Merge reconstructs the merged variant by creating a single low-rank update by reintroducing the (block-diagonal) singular values, and applying weighted addition:

$$\Delta W_{\text{TSV-M}} = U_\perp \Sigma V_\perp^\top, \qquad W_{\text{merged}} = W_0 + \lambda \Delta W_{\text{TSV-M}}. \tag{8}$$

Conceptually, TSV-Merge is a subspace-alignment mechanism: it compresses each task into its principal singular directions, aligns those directions across tasks to enforce mutual independence, and fuses them through a single low-rank reconstruction. The truncation regulates signal-noise trade-offs, Procrustes removes inter-task overlap, and the final scaling tunes how far the merged model moves from the base.

**Iso-C.** Iso-C (Marczak et al., 2025) introduces an isotropic model merging method designed to improve subspace alignment across task updates by flattening their singular value spectrum. Starting from the cumulative task vector $\Delta W_{\text{TA}}$ obtained via Task Arithmetic (Eq. (2)), Iso-C performs the following operation layerwise (we suppress the layer index $\ell$ for brevity). It computes an SVD

$$\Delta W_{\text{TA}} = U \Sigma V^\top, \qquad \Sigma = \text{diag}(\sigma_1, \ldots, \sigma_r), \tag{9}$$

where $\Sigma$ contains the singular values and $r$ denotes the effective rank. Rather than retaining the original (typically skewed) singular value distribution, which may overemphasize a few dominant task directions,

Iso-C replaces all singular values with their mean to enforce isotropy: $\bar{\sigma} = \frac{1}{r}\sum_{i=1}^{r}\sigma_i$ and $\Sigma_{\text{iso}} = \bar{\sigma}I_r$. The isotropically rescaled update and merged variant is then reconstructed as:

$$\Delta W_{\text{Iso-C}} = U\Sigma_{\text{iso}}V^{\top}, \quad W_{\text{merged}} = W_0 + \lambda\Delta W_{\text{Iso-C}}. \tag{10}$$

This operation equalizes the contribution of each principal direction, yielding a more balanced representation of task information. Conceptually, Iso-C can be viewed as a spectrum-flattened extension of Task Arithmetic: it preserves the same subspace spanned by $\Delta W_{\text{TA}}$ while imposing uniform scaling of its singular values.

**Subspace Boosting.** Subspace Boosting (Skorobogat et al., 2025) counteracts *rank collapse*, i.e. the tendency of merged task vectors to compress variance into a few dominant singular directions as multiple fine-tuned variants are combined. The method is applied layerwise; for clarity, we suppress the layer index $\ell$ throughout. Subspace Boosting performs an SVD of the merged update ($\Delta W = U\Sigma V^{\top}$), where the diagonal entries of $\Sigma = \text{diag}(\sigma_1, \ldots, \sigma_r)$ represent the energy of the corresponding subspace directions. The cumulative normalized energy is computed as $n_j = \frac{\sum_{i \leq j}\sigma_i}{\sum_{i=1}^{r}\sigma_i}$, and a boosting threshold $\beta$ determines the spectral cutoff index $j^* = \min\{j : n_j \geq \beta\}$. Singular values beyond this threshold are elevated to the cutoff value $\sigma_{j^*}$, producing a flattened spectrum. The boosted update is then constructed as

$$\Delta W_{\text{boosted}} = U\Sigma^{\star}V^{\top}, \qquad \sigma_j^{\star} = \begin{cases} \sigma_j, & j \leq j^*, \\ \sigma_{j^*}, & j > j^*, \end{cases} \qquad W_{\text{merged}} = W_0 + \lambda\Delta W_{\text{boosted}}. \tag{11}$$

Conceptually, Subspace Boosting broadens the effective subspace spanned by the merged variant by redistributing energy from dominant to weaker singular directions. The method is agnostic to the underlying merging strategy and can be seamlessly applied to any task-vector-based approach, such as Task Arithmetic or TIES-Merging.

### 4.2 Experimental Setup and Results

**Experimental Setup.** Apart from the merging algorithms, our setup mirrors Section 3. We integrated all available implementations into the *mergekit* library to provide a single, unified pipeline. We reuse the same base models as in Section 3, the same 12 checkpoints per base model, the identical subset-sampling over merge sizes, and the same evaluation configuration to isolate the effect of the merging algorithm itself. Following our ablation studies (Appendix C), we set $\lambda = 0.1$ for TIES+SB and $\lambda = 1.0$ for all other methods, while fixing the Subspace Boosting threshold to $\beta = 0.2$.

**Results.** Fig. 6 shows the average performance across benchmarks for all merged models and subspace merging methods. Trends for the different methods are consistent across LLMs: Both TSV-Merge and Iso-C exhibit steady declines in average accuracy as the number of merged models increases, indicating that their dimensional truncation and orthogonalization operations progressively discard informative components when aggregating multiple checkpoints. In contrast, methods utilizing Subspace Boosting avoid this degradation. TIES + SB demonstrates a highly stable profile, consistently remaining slightly above the base model, though it does not achieve significant scaling behavior. TA + SB exhibits high variance at small merge sizes but improves steadily with scale, eventually matching or even surpassing the base model's accuracy at large $n$, at a similar level to the original Task Arithmetic. These results indicate that subspace projection and flattening generally do not produce constructive interference in LLMs, whereas Task Arithmetic paired with Subspace Boosting remains the only setup that benefits from scaling the number of experts. However, given that this trend mirrors pure Task Arithmetic (see Fig. 3), this is mostly due to TA, while Subspace Boosting is not harmful here.

In Table 3, we again quantify these trends by reporting the probability of surpassing the base model and the average relative improvement across merge sizes. TA + SB consistently transitions from unstable early performance to strong, near-100% success as $n$ increases. At small merge sizes, success rates remain low—23% at $n=2$ with an average relative change of $-4.52$—but they rise steadily to 98% at $n=10$ and reach 100% at $n=12$, with corresponding improvements of $+0.89$ and $+1.07$. Also, TIES + SB improves from 72% success ($n=2$) to 100% ($n=12$), but with capped relative gains ($+0.06$ to $+0.36$). In contrast, TSV-Merge

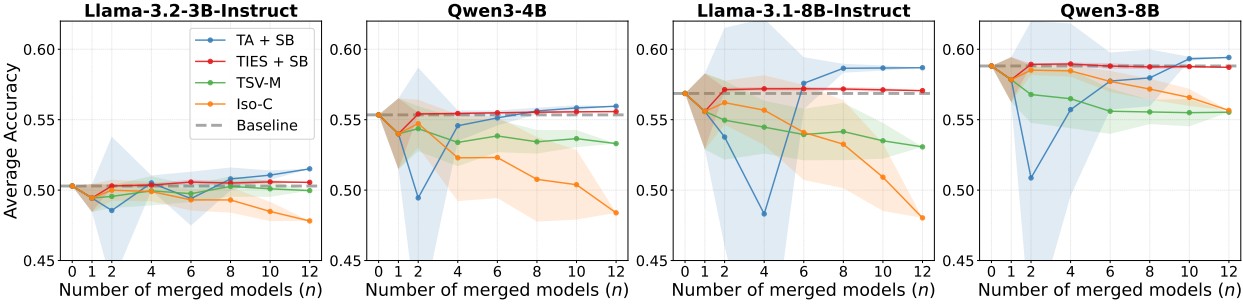

Figure 6: Average accuracy and standard deviation of the models across all benchmarks. From left to right, models are LLAMA 3.2 3B, QWEN3 4B, LLAMA 3.1 8B, QWEN3 8B, respectively. Shaded areas indicate the standard deviation over different samples of merged checkpoints.

| Model | Method | Base | $n$=1 (12) | $n$=2 (15) | $n$=4 (15) | $n$=6 (15) | $n$=8 (15) | $n$=10 (15) | $n$=12 (1) |
|---|---|---|---|---|---|---|---|---|---|
| LLAMA-3B | TA + SB | 50.3 | 17 / -0.85 | 13 / -1.74 | 67 / +0.22 | 60 / -0.90 | 87 / +0.50 | 100 / +0.77 | 100 / +1.22 |
| | TIES + SB | 50.3 | 17 / -0.85 | 47 / +0.01 | 47 / +0.07 | 87 / +0.28 | 93 / +0.21 | 100 / +0.29 | 100 / +0.26 |
| | TSV-M | 50.3 | 17 / -0.85 | 27 / -0.74 | 47 / -0.34 | 27 / -0.54 | 47 / -0.03 | 27 / -0.20 | 0 / -0.33 |
| | Iso-C | 50.3 | 17 / -0.85 | 33 / -0.30 | 47 / -0.39 | 20 / -0.98 | 13 / -0.99 | 0 / -1.81 | 0 / -2.48 |
| LLAMA-8B | TA + SB | 56.9 | 25 / -1.28 | 27 / -3.10 | 67 / -8.55 | 87 / +0.72 | 100 / +1.79 | 100 / +1.81 | 100 / +1.83 |
| | TIES + SB | 56.9 | 25 / -1.28 | 100 / +0.27 | 100 / +0.33 | 100 / +0.33 | 100 / +0.31 | 100 / +0.25 | 100 / +0.20 |
| | TSV-M | 56.9 | 25 / -1.28 | 33 / -1.90 | 20 / -2.39 | 7 / -2.92 | 13 / -2.70 | 0 / -3.36 | 0 / -3.79 |
| | Iso-C | 56.9 | 25 / -1.28 | 53 / -0.65 | 40 / -1.19 | 27 / -2.77 | 20 / -3.60 | 0 / -5.94 | 0 / -8.84 |
| QWEN-4B | TA + SB | 55.3 | 25 / -1.34 | 33 / -5.90 | 33 / -0.77 | 47 / -0.22 | 87 / +0.28 | 93 / +0.49 | 100 / +0.61 |
| | TIES + SB | 55.3 | 25 / -1.34 | 73 / +0.06 | 87 / +0.09 | 100 / +0.14 | 100 / +0.19 | 100 / +0.21 | 100 / +0.23 |
| | TSV-M | 55.3 | 25 / -1.34 | 13 / -0.98 | 7 / -1.95 | 0 / -1.50 | 7 / -1.91 | 0 / -1.69 | 0 / -2.04 |
| | Iso-C | 55.3 | 25 / -1.34 | 27 / -0.62 | 7 / -3.05 | 0 / -3.03 | 7 / -4.58 | 0 / -4.96 | 0 / -6.95 |
| QWEN-8B | TA + SB | 58.8 | 33 / -0.97 | 20 / -7.95 | 67 / -3.10 | 60 / -1.07 | 67 / -0.84 | 100 / +0.51 | 100 / +0.60 |
| | TIES + SB | 58.8 | 33 / -0.97 | 66 / -0.09 | 100 / +0.38 | 100 / +0.49 | 100 / +0.56 | 100 / +0.66 | 100 / +0.73 |
| | TSV-M | 58.8 | 33 / -0.97 | 13 / -2.03 | 13 / -2.33 | 0 / -3.21 | 0 / -3.26 | 0 / -3.32 | 0 / -3.28 |
| | Iso-C | 58.8 | 33 / -0.97 | 20 / -0.30 | 27 / -0.35 | 0 / -1.10 | 0 / -1.64 | 0 / -2.23 | 0 / -3.16 |
| **Average** | TA + SB | 55.3 | 25 / -1.11 | 23 / -4.52 | 58 / -2.55 | 63 / -0.37 | 86 / +0.43 | 98 / +0.89 | 100 / +1.07 |
| | TIES + SB | 55.3 | 25 / -1.11 | 72 / +0.06 | 84 / +0.22 | 97 / +0.31 | 98 / +0.32 | 100 / +0.35 | 100 / +0.36 |
| | TSV-M | 55.3 | 25 / -1.11 | 22 / -1.41 | 22 / -1.75 | 8 / -2.04 | 17 / -2.00 | 7 / -2.14 | 0 / -2.36 |
| | Iso-C | 55.3 | 25 / -1.11 | 33 / -0.47 | 30 / -1.23 | 12 / -1.97 | 10 / -2.70 | 0 / -3.73 | 0 / -5.36 |

Table 3: Constructive interference results for Subspace–based merging methods across models. Each entry contains two quantities: the percentage of merge combinations that exceed the base model's accuracy, and the mean relative accuracy improvement for those combinations. Column headers use the notation $n = m\,(k)$, where $n$ is the number of models merged and $k$ is the number of evaluated merge combinations for that value of $n$. Base indicates base model accuracy.

and Iso-C deteriorate monotonically in both probability and relative improvement as the number of experts grows. TSV-Merge decreases from 22% success and $-1.41$ at $n$=2 to 0% and $-2.36$ at $n$=12, while Iso-C moves from 33% and $-0.47$ at $n$=2 to 0% and $-5.36$ at $n$=12. On average, subspace projection–based methods suppress rather than exploit beneficial diversity, whereas Task Arithmetic with Subspace Boosting remains the only configuration whose performance scales constructively with increasing model diversity.

## 5  Discussion and Limitations

**Why Merging Methods Underperform in "In-the-Wild" Scenarios.** Subspace-based merging methods rely on strong assumptions about the geometry of fine-tuned checkpoints, which typically hold when models specialize on distinct, well-defined tasks. In such settings, coherent update directions enable operations like SVD truncation, orthogonalization, or isotropization to align or reshape task subspaces constructively. In our setup, however, we merge randomly sampled checkpoints, which also is practically relevant and directly evaluates the promise of merging methods of reusing the vast repository of publicly available model variants (Ramé et al., 2023a). Their update directions need not form stable subspaces and may conflict substantially with each other. Consequently, subspace transformations can distort the combined update and push the merged model outside the linearly mode-connected region around the base LLM, increasing the risk of performance degradation.

In contrast, Task Arithmetic makes no subspace assumptions and effectively averages task vectors. When checkpoints are diverse, this averaging remains close to the base model, yielding modest but consistently positive gains. This explains why Task Arithmetic is more successful under random sampling, whereas subspace-based methods, though effective in their intended regimes, often underperform in our setting.

**Homogeneous vs. Heterogeneous Merging.** To further isolate the source of interference, we analyzed homogeneous merges in Appendix E, where experts were drawn from distinct, non-conflicting domains (Mathematics and Medicine). Unlike the random "in-the-wild" sampling, we found that merging experts from these clearly defined domains, even when combining both domains simultaneously, resulted in negligible performance loss compared to domain-specific merges. For instance, a joint model merging both Math and Medical experts performed nearly identically to specialized merges on their respective tasks. This contrast confirms that the performance degradation observed in our main evaluation stems specifically from the unstructured heterogeneity of randomly sampled checkpoints. When experts possess conflicting update directions or high variance without clear task separation, advanced merging mechanisms struggle to extract useful signals, whereas they succeed when task roles are distinct.

**Limitations and Future Directions.** While our evaluation is extensive, it is not exhaustive. First, we intentionally focused on LLMs and did not evaluate encoder-decoder or multimodal models, where subspace geometry and fine-tuning dynamics may differ. Second, our experimental design omits pre-merging alignment or clustering steps to isolate intrinsic effects of merging methods. Future work should investigate whether pre-merging strategies like spectral filtering of task vectors or clustering improve performance.

## 6  Conclusion

We present a large-scale study of "in-the-wild" model merging for LLMs. Across four model families, twelve fine-tuned checkpoints per base model, and sixteen benchmarks, we find that only Task Arithmetic reliably produces constructive interference, i.e., improving upon both the base model and all individual checkpoints. In contrast, interference-aware and subspace-based approaches (TIES-Merging, Model Stock, TSV-Merge, Iso-C, Subspace Boosting) fail to provide gains and degrade performance when not properly tuned.

These findings suggest that it is difficult, but not impossible, to improve the base model by simply merging a heterogeneous set of fine-tuned versions and outperforming every checkpoint involved. Additionally, we find that Task Arithmetic yields better results on this task, while more sophisticated methods, such as TIES or subspace-based methods, do not successfully extract knowledge from heterogeneous checkpoints to improve base model performance.

A priority for future work is designing merging algorithms tailored to LLMs and validating them directly in this setting rather than relying solely on image-classification benchmarks. Our implementation, which combines *mergekit* with *lm-eval-harness*, provides a standardized framework for such evaluations. Finally, merging-aware fine-tuning, which explicitly encourages complementary specializations, may further amplify the benefits of model merging, as our results with arbitrary checkpoints already suggest its potential.

**Broader Impact Statement**

This work investigates the reliability and limitations of model merging techniques for large language models. By clarifying when constructive interference occurs, our findings can help practitioners combine fine-tuned models more efficiently, potentially reducing computational cost and energy consumption associated with retraining. The study may also support open research by enabling reuse of publicly available fine-tuned checkpoints.

At the same time, model merging raises ethical and practical concerns. Automatically combining models without understanding their data provenance or domain biases can amplify undesirable behaviors, privacy risks, or misinformation learned from individual experts. Our results highlight that merging is not universally reliable and should be applied cautiously, with careful monitoring of model behavior and documentation of merged checkpoints. Overall, we believe that greater transparency and empirical rigor in evaluating merging methods contributes positively to responsible large-model development.

**Acknowledgements**

This work was partially funded by the ERC (853489 - DEXIM) and the Alfried Krupp von Bohlen und Halbach Foundation, for which we thank them for their generous support. The authors gratefully acknowledge the scientific support and resources of the AI service infrastructure *LRZ AI Systems* provided by the Leibniz Supercomputing Centre (LRZ) of the Bavarian Academy of Sciences and Humanities (BAdW), funded by Bayerisches Staatsministerium für Wissenschaft und Kunst (StMWK).

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

# Supplementary Material

## A    Fine-tuned Checkpoints

For each base model, we used 12 publicly available fine-tuned checkpoints from the Hugging Face Hub. The complete list is provided below for reproducibility.

**meta-llama/Llama-3.2-3B-Instruct**

1. MergeBench/Llama-3.2-3B-Instruct_instruction
2. MergeBench/Llama-3.2-3B-Instruct_multilingual
3. MergeBench/Llama-3.2-3B-Instruct_math
4. MergeBench/Llama-3.2-3B-Instruct_coding
5. MergeBench/Llama-3.2-3B-Instruct_safety
6. belyakoff/llama-3.2-3b-instruct-fine-tuned
7. jjzha/Llama-3.2-3B-Instruct-SEFL
8. acon96/Home-Llama-3.2-3B
9. dolphinium/Llama-3.2-3B-instruct-fine-tuned-model
10. FuseAI/FuseChat-Llama-3.2-3B-Instruct
11. VaidikML0508/Shark-Tank-Offer-Evaluator-llama3.2-3B-Instruct-GRPO-16bits-V1
12. huihui-ai/Llama-3.2-3B-Instruct-abliterated

**meta-llama/Llama-3.1-8B-Instruct**

1. mims-harvard/TxAgent-T1-Llama-3.1-8B
2. arcee-ai/Llama-3.1-SuperNova-Lite
3. DeepMount00/Llama-3.1-8b-ITA
4. mlabonne/Meta-Llama-3.1-8B-Instruct-abliterated
5. Kukedlc/NeuralLLaMa-3-8b-ORPO-v0.3
6. curiositytech/MARS-v0.2
7. SentientAGI/Dobby-Mini-Unhinged-Llama-3.1-8B
8. UW-Madison-Lee-Lab/Llama-PRM800K
9. barc0/Llama-3.1-ARC-Potpourri-Induction-8B
10. AIDX-ktds/ktdsbaseLM-v0.13-onbased-llama3.1
11. TheFinAI/Fino1-8B
12. tokyotech-llm/Llama-3.1-Swallow-8B-v0.5

**Qwen/Qwen3-4B**

1. mlxha/Qwen3-4B-grpo-medmcqa
2. Menlo/Jan-nano
3. Vikhrmodels/QVikhr-3-4B-Instruction
4. POLARIS-Project/Polaris-4B-Preview
5. mlabonne/Qwen3-4B-abliterated
6. ValiantLabs/Qwen3-4B-Esper3
7. KissanAI/ThinkingDhenu1-CRSA-India-preview
8. russwest404/Qwen3-4B-ReTool-SFT
9. Intelligent-Internet/II-Search-4B
10. Dev9124/qwen3-finance-model
11. qihoo360/Light-IF-4B
12. prithivMLmods/Draconis-Qwen3_Math-4B-Preview

**Qwen/Qwen3-8B**

1. Trendyol/Trendyol-LLM-8B-T1
2. huihui-ai/Huihui-Qwen3-8B-abliterated-v2
3. ValiantLabs/Qwen3-8B-Esper3
4. miromind-ai/MiroThinker-8B-SFT-v0.1
5. Goedel-LM/Goedel-Prover-V2-8B
6. mlabonne/Qwen3-8B-abliterated
7. soob3123/GrayLine-Qwen3-8B
8. TheFinAI/Fin-o1-8B
9. AXCXEPT/Qwen3-EZO-8B-beta
10. tomg-group-umd/DynaGuard-8B
11. NoemaResearch/Apollo-1-8B
12. Vikhrmodels/QVikhr-3-8B-Instruction

## B    Taskwise Accuracy of Models

In Figs. 7 to 10, we provide detailed task-wise performance breakdowns for all evaluated base models. Across all model families and sizes, we observe consistent behavioral patterns that align with the aggregated results reported in the main text. Specifically, Task Arithmetic and its subspace-boosted variant demonstrate robust scaling, maintaining or improving accuracy on diverse benchmarks such as `arc_challenge` and `winogrande` as $n$ increases. In contrast, Iso-C and TSV-M suffer from performance degradation on knowledge-intensive and reasoning tasks like `medmcqa` and `mmlu`, particularly as the number of merged checkpoints grows. Model Stock, TIES-Merging, and its subspace-boosted variant rarely deviates significantly from the base model's performance profile. These task-level visualizations confirm that the superior average performance of Task Arithmetic is driven by consistent gains across a wide range of evaluation dimensions rather than outliers in specific tasks.

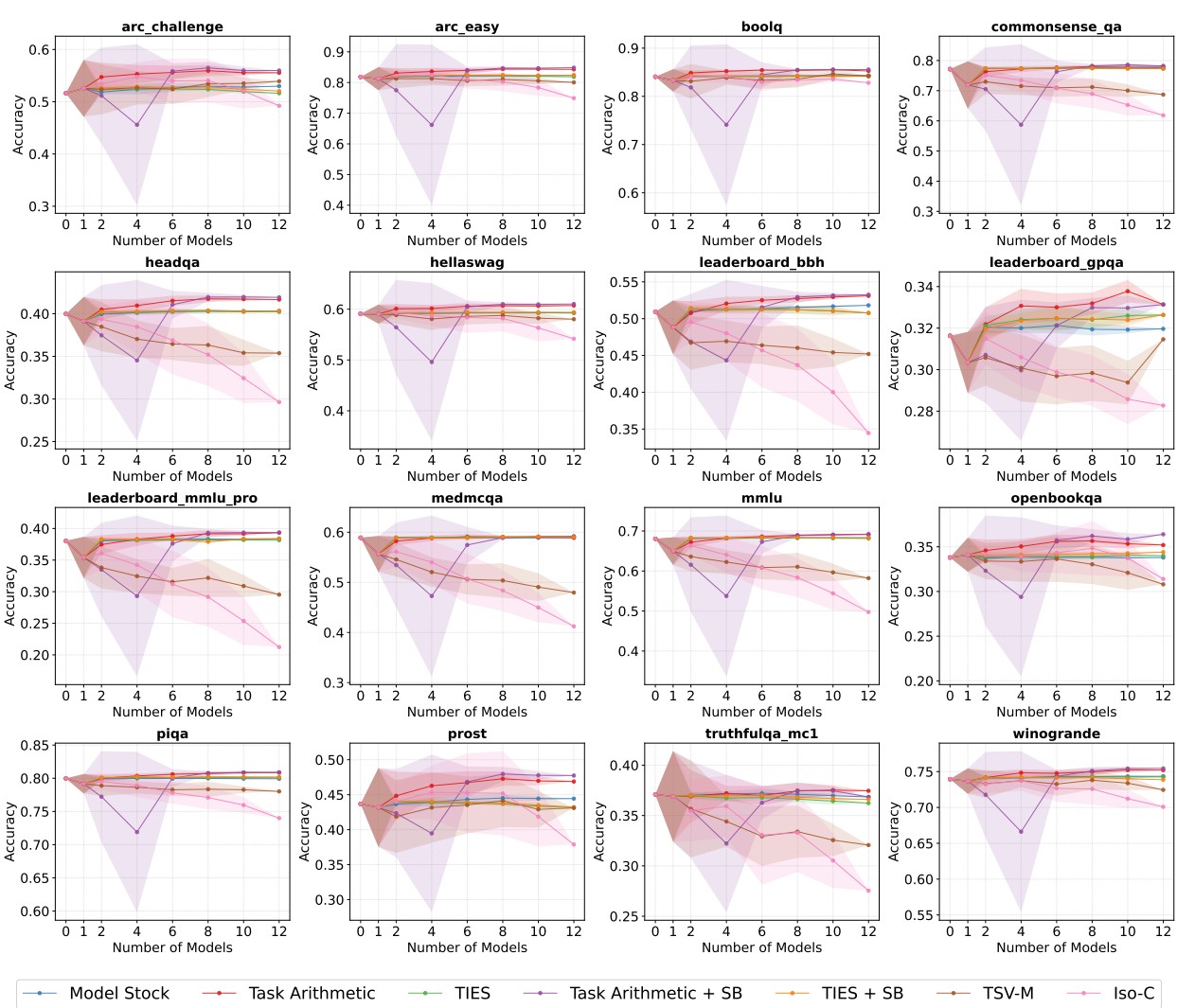

Figure 7: Taskwise accuracy and standard deviation of LLAMA 3.1 8B.

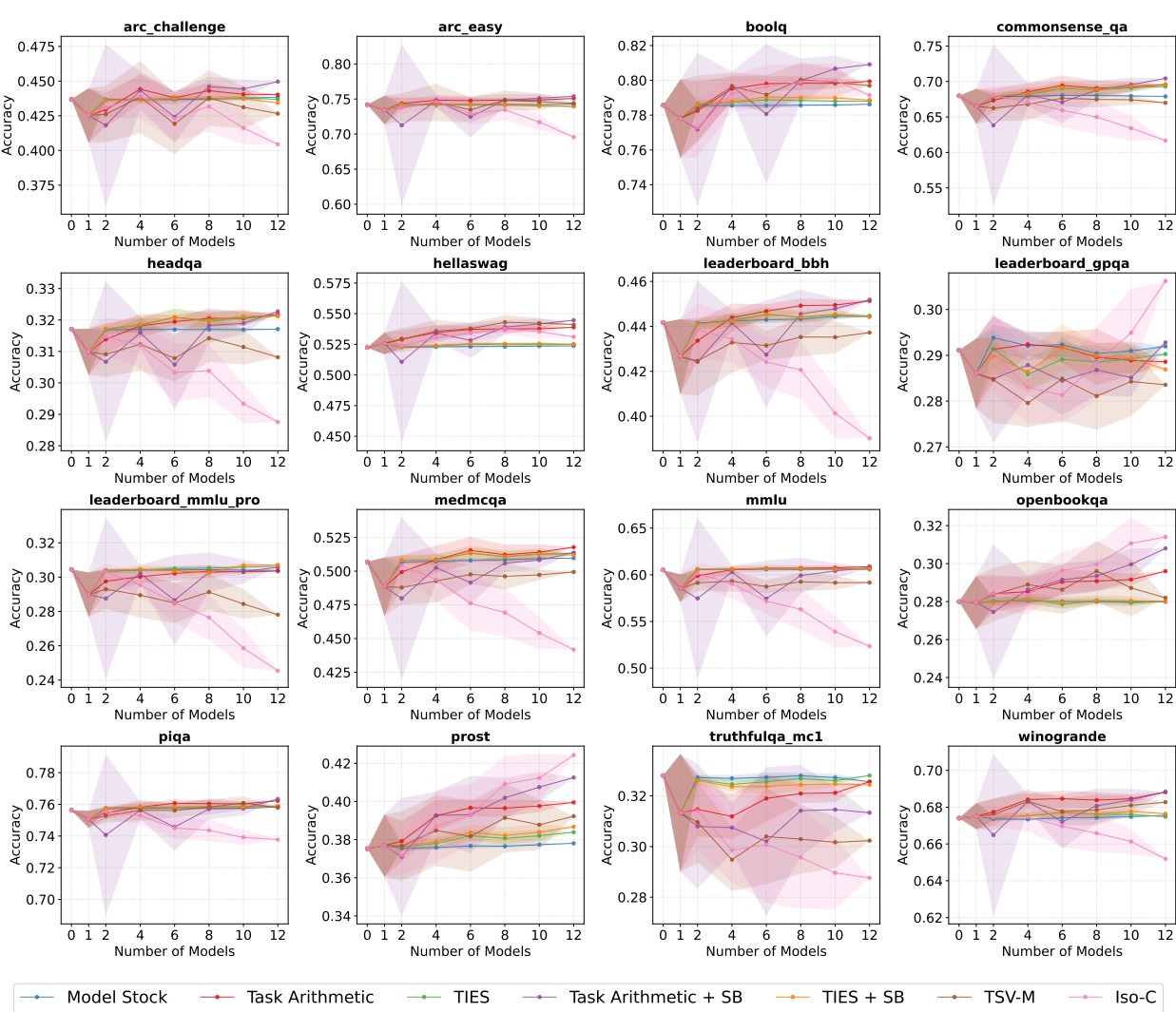

Figure 8: Taskwise accuracy and standard deviation of LLAMA 3.2 3B.

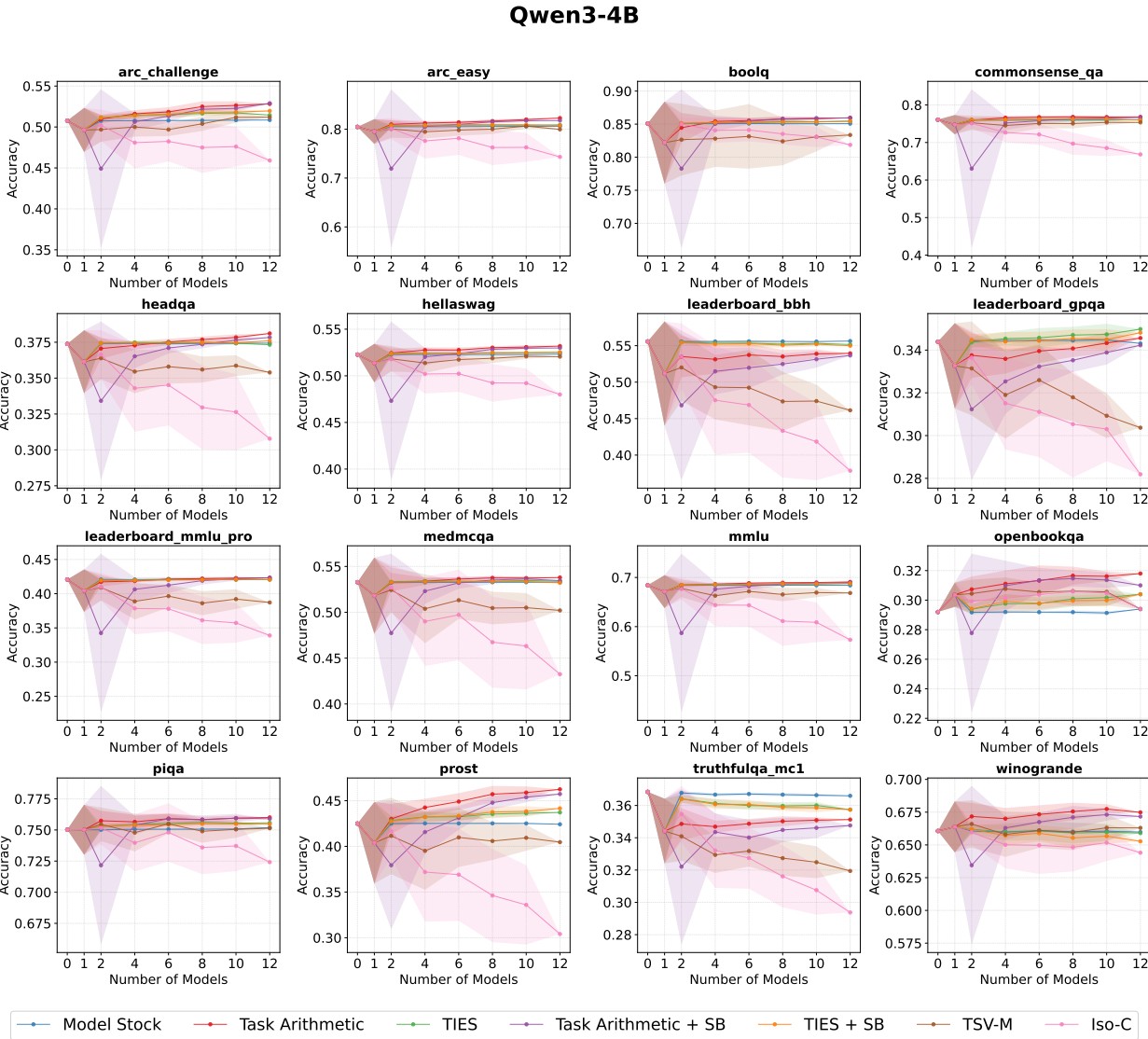

Figure 9: Taskwise accuracy and standard deviation of QWEN3 4B.

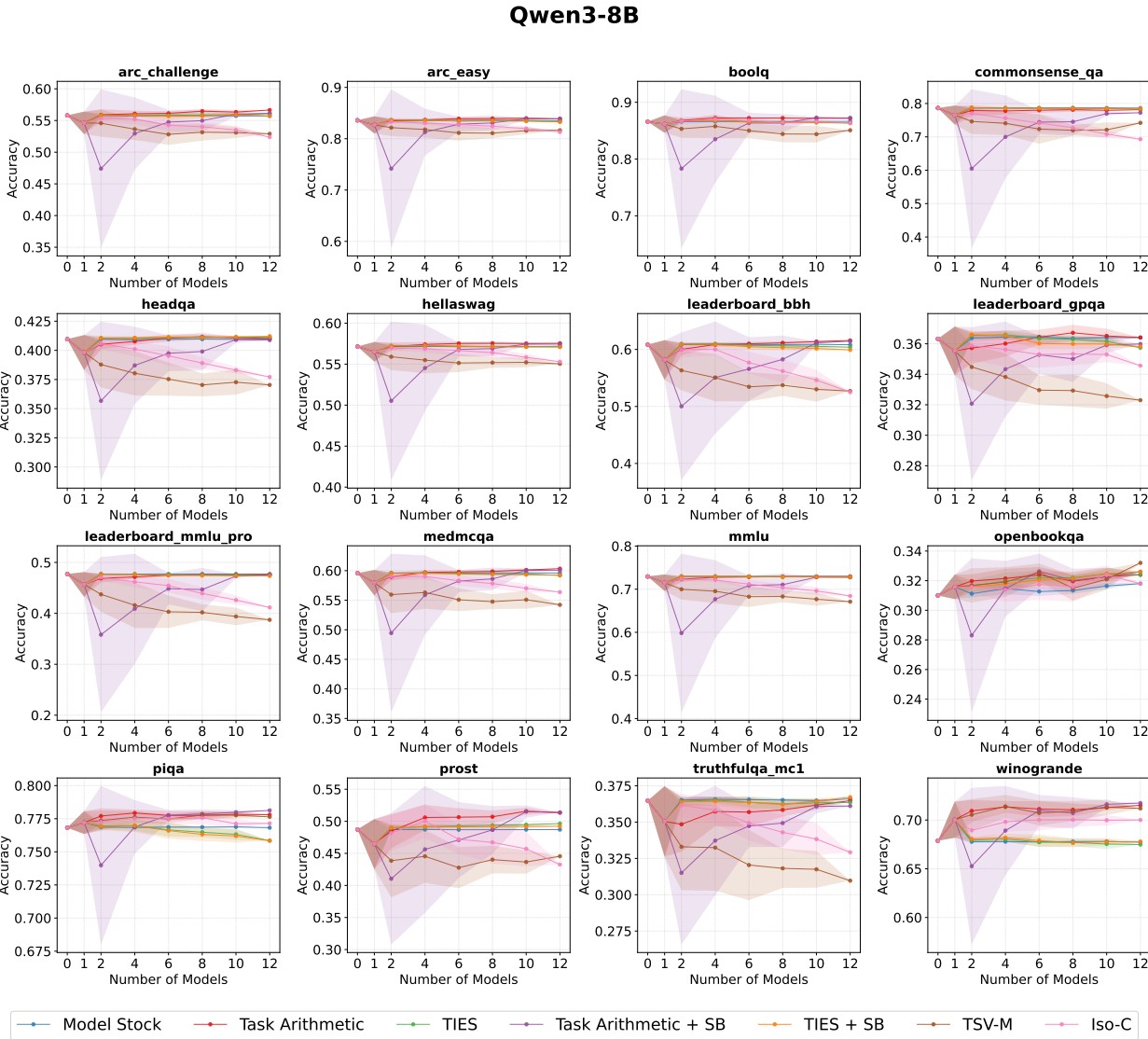

Figure 10: Taskwise accuracy and standard deviation of Qwen3 8B.

# C   Hyperparameter Ablations and Sensitivity Analysis

In this section, we analyze the sensitivity of the merging methods to their key hyperparameters: the scaling coefficient $\lambda$, the pruning density $k$ (for TIES-Merging), and the spectral threshold $\beta$ (for Subspace Boosting). We specifically investigate the structural reasons why Task Arithmetic and TIES-Merging exhibit divergent behaviors regarding the scaling coefficient $\lambda$ when applied to heterogeneous, "in-the-wild" checkpoints. Unless otherwise stated, all ablations are performed by merging all 12 fine-tuned variants and sweeping the corresponding method-specific hyperparameters. We report average accuracy over the evaluation suite.

## C.1   Task Arithmetic: Stability via Normalization and Cancellation

In Fig. 11, we analyze the sensitivity of Task Arithmetic to the scaling coefficient $\lambda$. We initially observed a flat performance curve across the standard range $\lambda \in [0.1, 1.9]$. To test whether this stability holds indefinitely or is merely a matter of scale, we extended the sweep to $\lambda = 10$. As shown in the rightmost portion of the plot, performance drops significantly at this extreme, confirming that the method is not invariant to scaling but rather highly robust within the typical hyperparameter range. We attribute this robustness to the normalization configuration employed by *mergekit*. By default, we consistently enabled normalization in our experiments, which controls how individual task vectors are aggregated. Formally, let $\delta_i$ be the task vector of model $i$ and $\alpha_i$ its scalar weight. With normalization enabled, the merged update $\Delta$ is computed as a weighted average:

$$\Delta_{\text{norm}} = \frac{\sum_i \alpha_i \, \delta_i}{\sum_i \alpha_i}. \tag{12}$$

As shown in Appendix D, the task vectors of the randomly selected checkpoints used in this study are largely orthogonal. Consequently, when averaged via normalization, the incoherent directions cancel out, resulting in a merged task vector with a very small magnitude. Because the base update vector is close to zero, scaling it by $\lambda \in [0.1, 1.9]$ results in negligible movement in parameter space, keeping the model within the low-loss basin of the base checkpoint. Only when $\lambda$ is pushed to extreme values (e.g., $\lambda = 10$) does the update magnitude become large enough to show significant performance change.

To validate that this stability is indeed an artifact of averaging-induced cancellation, we perform an ablation where normalization is disabled. In this setting, the merge becomes a weighted sum ($\Delta = \sum_i \alpha_i \delta_i$). Without the normalizing divisor, the heterogeneous updates accumulate magnitude rather than averaging out, yielding a task vector that is more likely to disrupt model performance. The results in Fig. 12 strongly support this analysis: in the absence of normalization, the model exhibits sensitivity to $\lambda$, with accuracy degrading rapidly as the scaling factor increases, even within the standard range.

## C.2   TIES-Merging: Sparsity Prevents Cancellation

For TIES-Merging, we perform a grid search over the pruning density $k$ (top-$k\%$) and the scaling factor $\lambda$. Figs. 13 to 16 illustrate the relationship between the $L_2$-norm of the merged task vector and average accuracy.

We observe two distinct trends that differentiate TIES from Task Arithmetic:

1. Unlike the flat curve of Task Arithmetic, TIES exhibits a monotonic degradation in accuracy as $\lambda$ increases from 0.1 to 1.9, regardless of the density setting.

2. Increasing $\lambda$ in TIES causes a rapid increase in the $L_2$-norm of the task vector, which correlates with performance degradation.

One might expect normalization to stabilize TIES just as it did for Task Arithmetic. However, our results show that while Task Arithmetic maintains stable performance across $\lambda \in [0.1, 1.9]$, TIES exhibits an increase in both the $L_2$-norm and the performance degradation as $\lambda$ grows (see Figs. 13 to 16). We hypothesize that

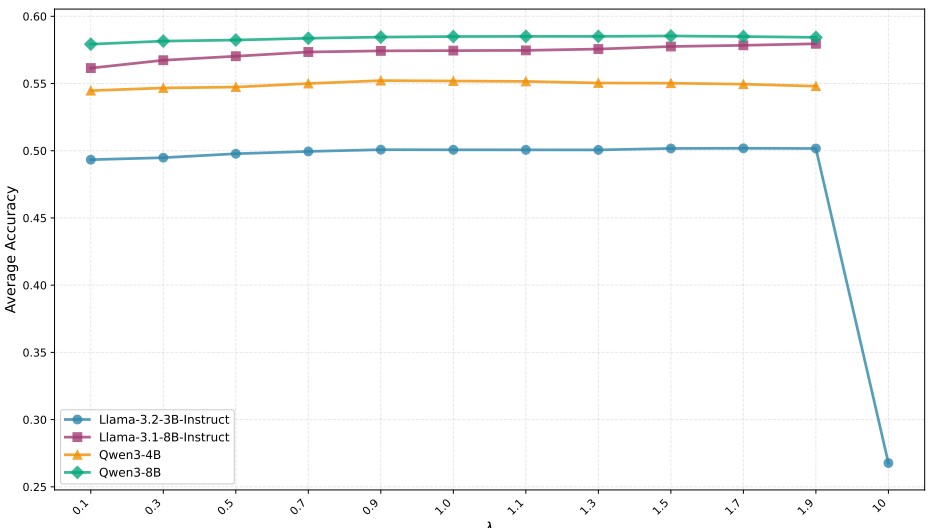

Figure 11: **Task Arithmetic: Effect of mixing coefficient** $\lambda$**.** We sweep the interpolation weight $\lambda$ used to combine task updates in Task Arithmetic.

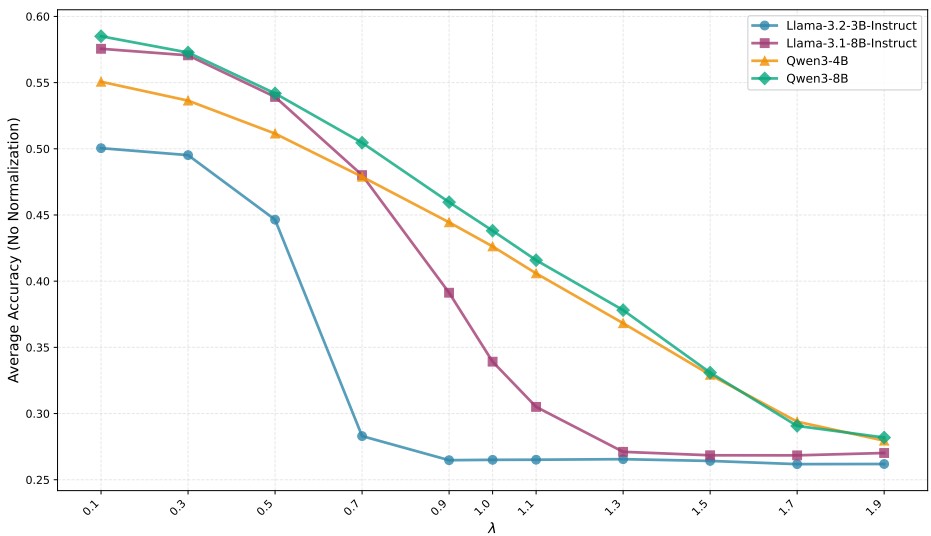

Figure 12: **Task Arithmetic (Without Normalization): Effect of mixing coefficient** $\lambda$**.** We sweep the interpolation weight $\lambda$ used to combine task updates in Task Arithmetic. Task Arithmetic is more sensitive to $\lambda$ scaling factor without normalization.

this sensitivity arises because the design of TIES structurally limits the cancellation effects that otherwise stabilize heterogeneous merges. Specifically:

- **Trimming:** By retaining only the top-$k\%$ magnitude parameters, TIES filters out the low-magnitude parameters that would typically facilitate averaging towards zero in a standard mean operation.

- **Sign Consensus:** By masking parameters that disagree on sign, TIES enforces directionality among the remaining weights.

In the context of random, heterogeneous checkpoints, we hypothesize that these mechanisms isolate high-magnitude parameters that, due to the lack of task alignment, do not encode a coherent shared skill but rather high-variance weights. Because these weights are forced into alignment by the consensus step, they accumulate rather than cancel out during the merge. This hypothesis is supported by the observed expansion of the task vector's $L_2$-norm in Figs. 13 to 16, which pushes the model far from the pre-trained weights even at moderate $\lambda$ values, resulting in the significant accuracy drops seen in our sweep.

Based on this analysis, we select $\lambda = 1.0$ for Task Arithmetic and $\lambda = 0.1$ for TIES in our main experiments.

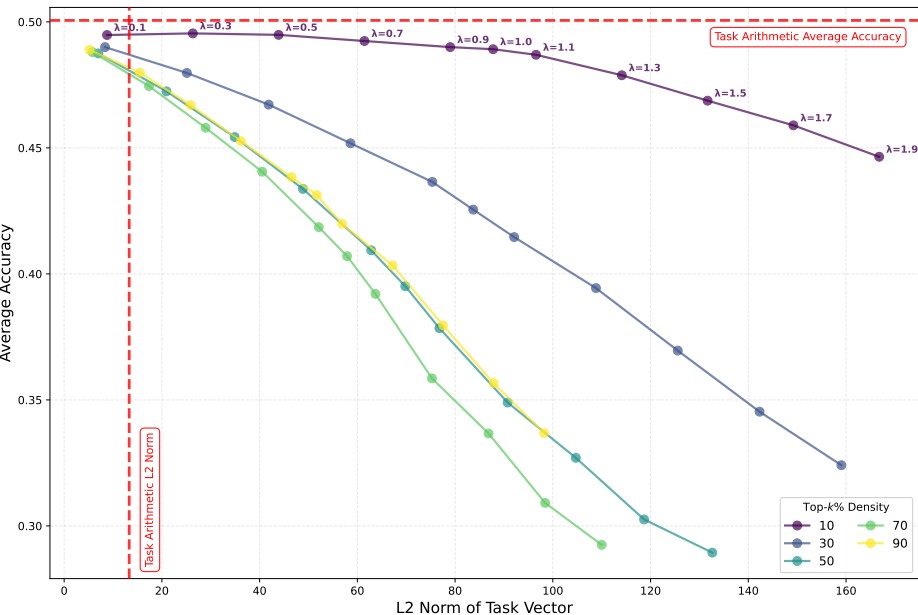

Figure 13: Average accuracy versus $L_2$-norm of the merged task vector for TIES on LLAMA 3.2 3B, under varying $\lambda$ and top-$k\%$ density settings.

## C.3  TIES top-$k\%$ Density

Fig. 17 illustrates the impact of the pruning density $k$ in TIES-Merging. The results reveal a distinct "U-shaped" trajectory: accuracy is maximized when the density is either very low or very high, while degrading significantly in the intermediate range. Although performance recovers as $k$ approaches 100%, we did not select this setting because at full density, the pruning mechanism is effectively disabled, making the method behaviorally nearly identical to standard Task Arithmetic. Therefore, to faithfully evaluate the sparsification properties that distinguish TIES-Merging from simple averaging, we selected the top-10% density for our main evaluation.

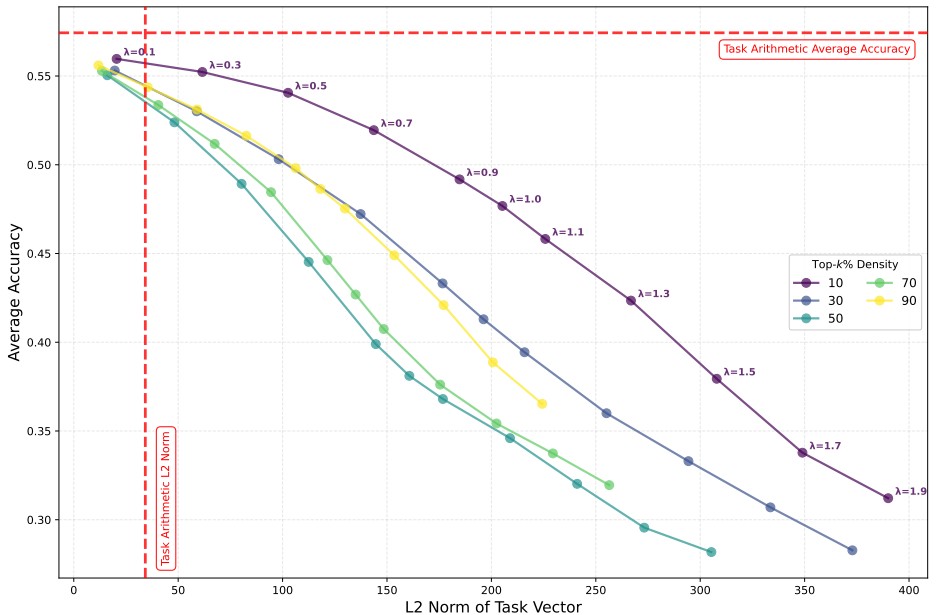

Figure 14: Average accuracy versus $L_2$-norm of the merged task vector for TIES on LLAMA 3.1 8B, under varying $\lambda$ and top-$k\%$ density settings.

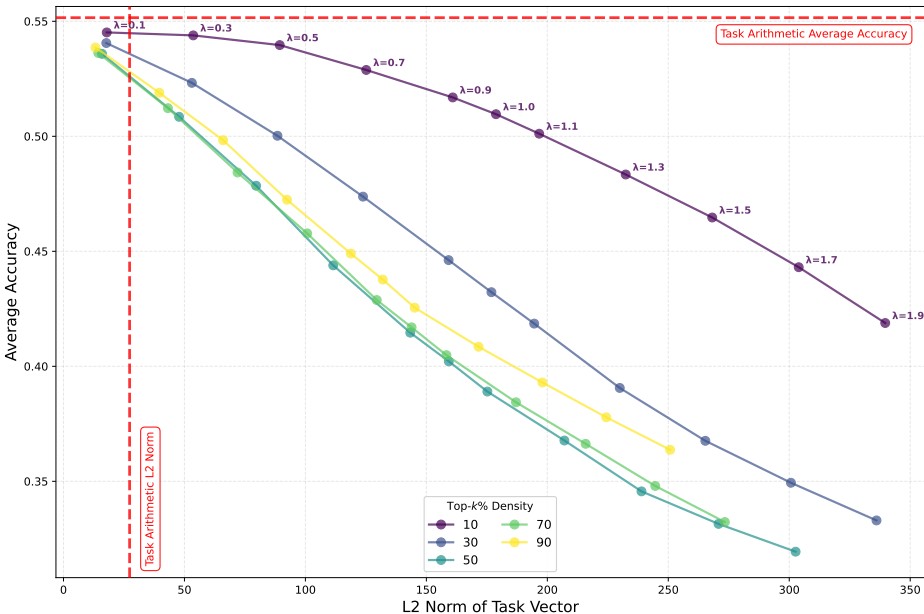

Figure 15: Average accuracy versus $L_2$-norm of the merged task vector for TIES on QWEN3 4B, under varying $\lambda$ and top-$k\%$ density settings.

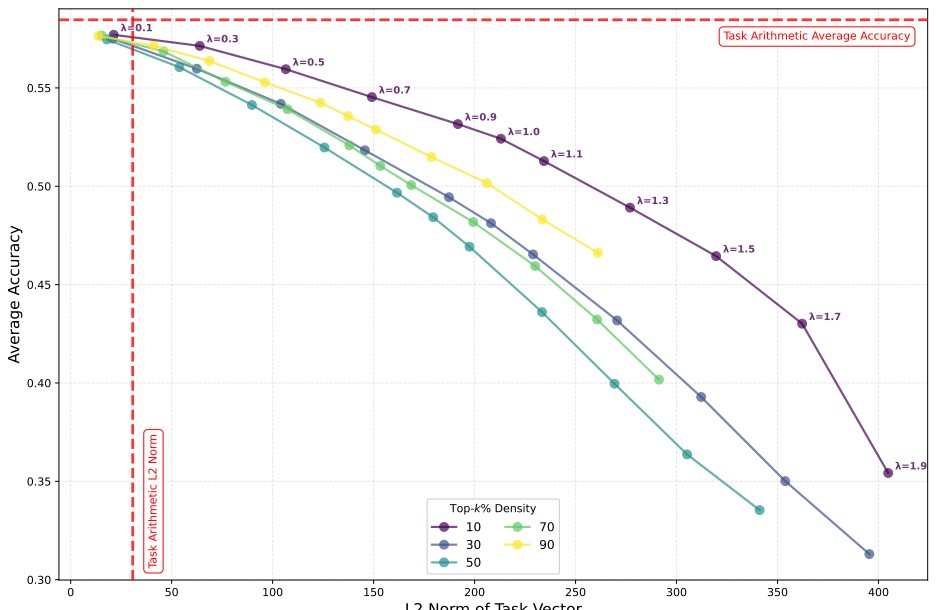

Figure 16: Average accuracy versus $L_2$-norm of the merged task vector for TIES on QWEN3 8B, under varying $\lambda$ and top-$k\%$ density settings.

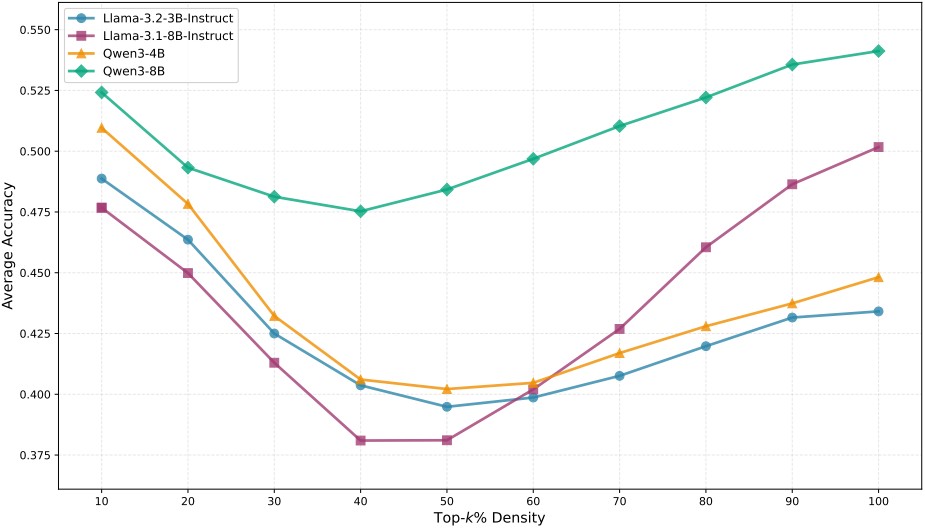

Figure 17: **TIES-Merging: effect of top-$k\%$ density.** We sweep the top-$k\%$ density, defined as retaining the top-$k\%$ largest-magnitude weights in TIES-Merging. Higher density (larger $k$) keeps more parameters active (less sparsity), whereas lower density (smaller $k$) enforces stronger sparsity.

### C.4 Subspace Boosting Threshold

Fig. 18 depicts the performance of Subspace Boosting as a function of the spectral threshold $\beta$. We observe a rapid performance gain as $\beta$ increases from 0 to 0.05, after which the accuracy stabilizes and remains robust across a wide range of values ($\beta \in [0.1, 0.5]$). This indicates that Subspace Boosting is not highly sensitive to the exact threshold, provided it is large enough. We therefore chose $\beta = 0.2$ for our experiments to ensure robust spectral flattening.

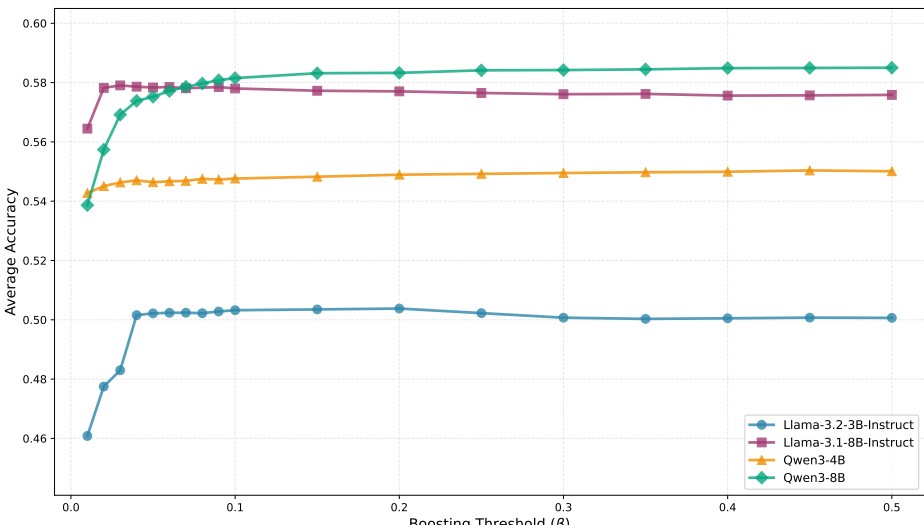

Figure 18: **Subspace Boosting: effect of the boosting threshold $\beta$.** We sweep the raw-proportion threshold $\beta \in [0, 1]$ in Subspace Boosting. For each SVD, singular values whose normalized cumulative sum is $\leq \beta$ are left unchanged; subsequent singular values are boosted by clamping them to the cutoff singular value. Accuracy vs. $\beta$ highlights how strengthening lower-energy directions mitigates interference and impacts overall performance.

## D   Pairwise Cosine Similarities of Task Vectors from Fine-Tuned Variants

Figs. 19 to 22 present the pairwise cosine similarities between task vectors obtained from the fine-tuned checkpoints used in this study. Across all model families, the cosine similarity values are concentrated near zero, indicating that the corresponding task vectors are largely orthogonal in parameter space. Importantly, each model family exhibits both positive and negative cosine similarity values, suggesting that while some fine-tuned variants induce mildly aligned task directions, others produce updates in opposing directions.

To complement these aggregate measurements, we additionally report layerwise sign heatmaps in Figs. 23 to 26. For each pair of fine-tuned models, cosine similarity is computed independently at each layer, and we record the number of layers exhibiting positive versus negative cosine similarity. These matrices therefore summarize the distribution of alignment signs across layers, rather than a single scalar similarity value. The layerwise analysis reveals that even when the global cosine similarity between two task vectors is close to zero, individual layers may induce aligned updates, opposing updates, or remain effectively neutral. This sign heterogeneity provides direct evidence for partial cancellation effects in Task Arithmetic, arising from mixtures of positively and negatively aligned layers rather than uniformly orthogonal updates.

Finally, across all base model families, we observe a non-negligible number of layers whose cosine similarity is equal to (or numerically indistinguishable from) zero. As a result, for a given model pair, the sum of positive and negative layer counts does not necessarily equal the total number of layers.

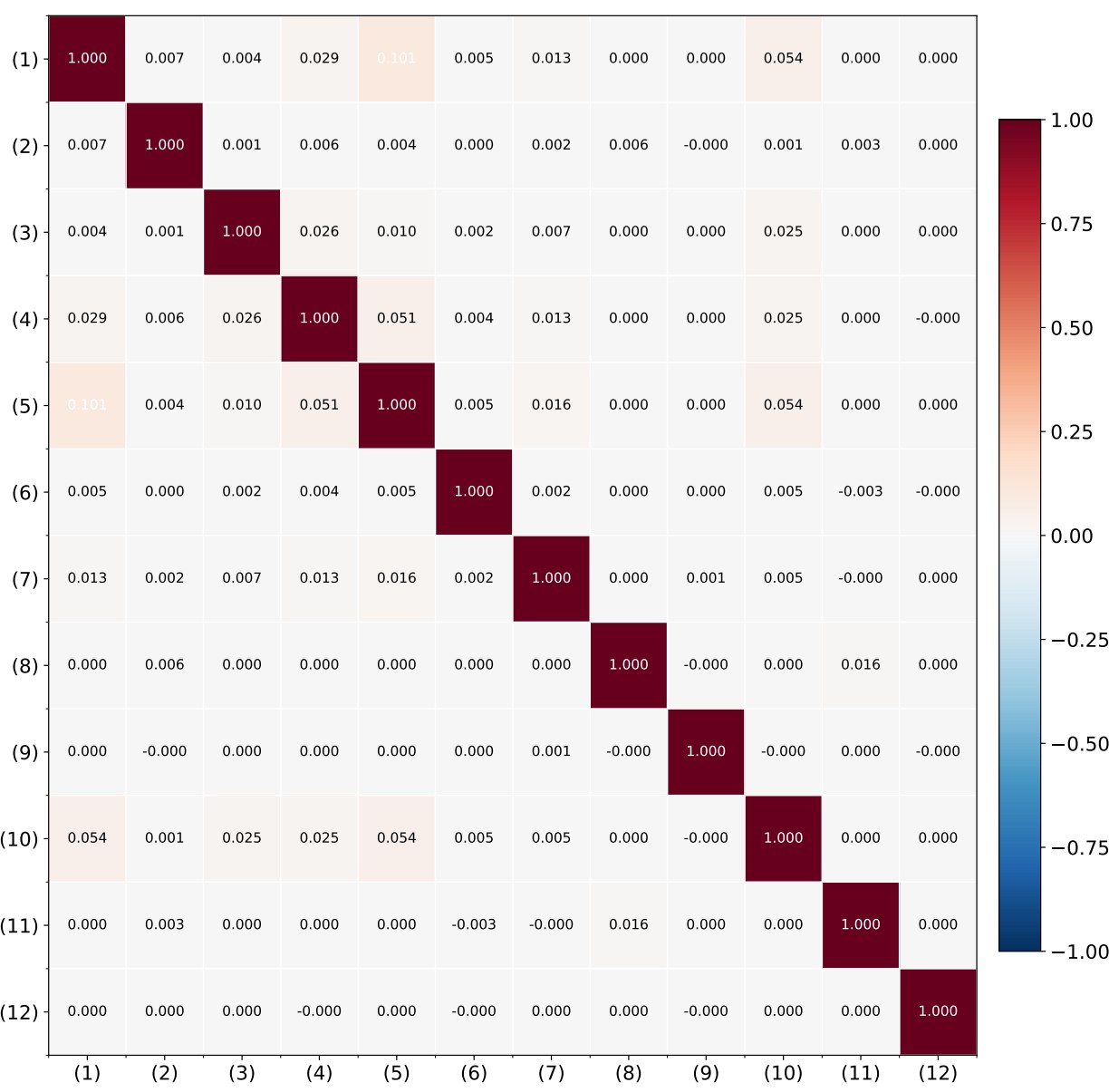

Figure 19: Pairwise cosine similarities between task vectors of fine-tuned LLAMA 3.2 3B variants. Indices 1–12 correspond to the checkpoint ordering listed in Appendix A.

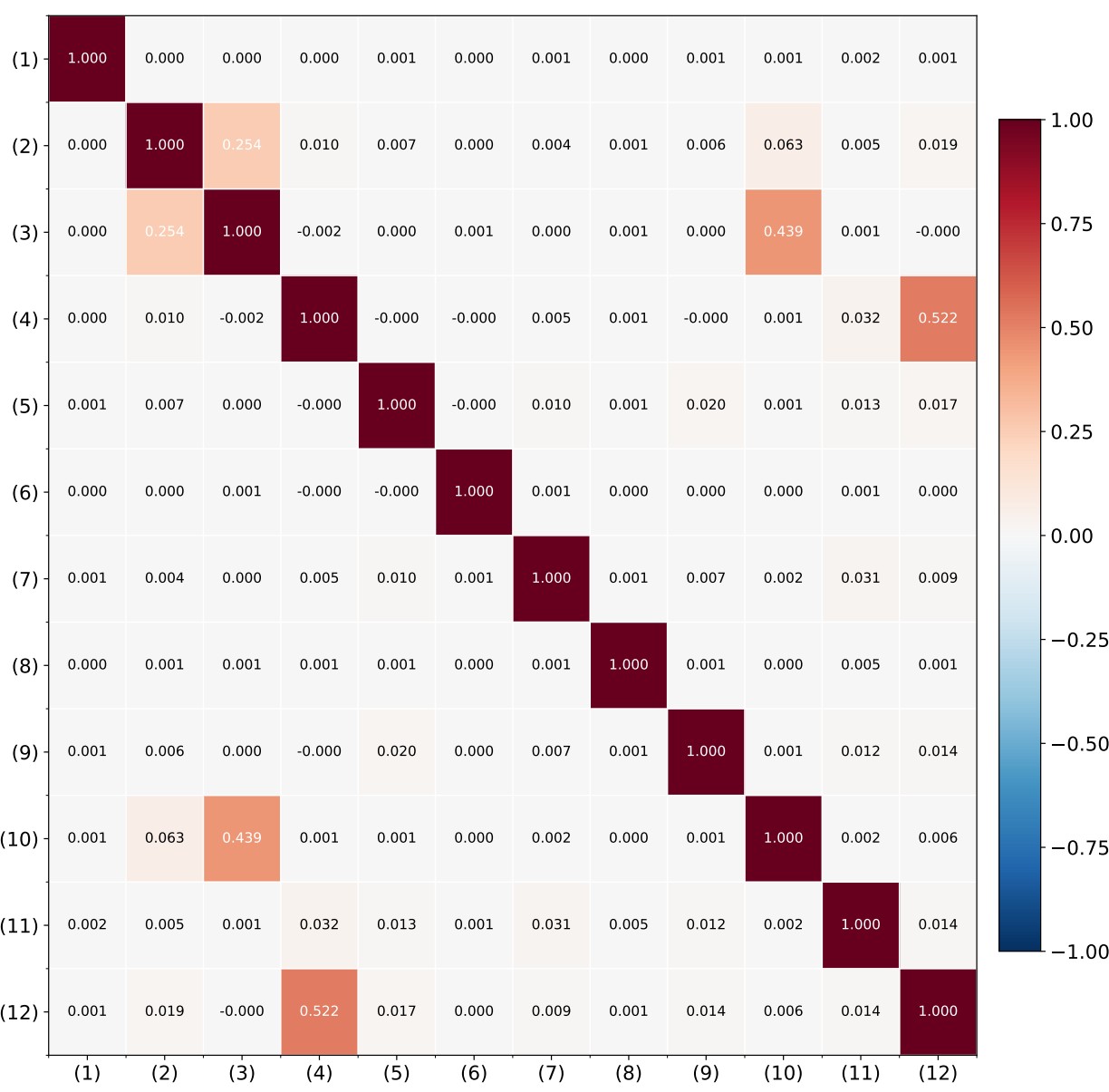

Figure 20: Pairwise cosine similarities between task vectors of fine-tuned LLAMA 3.1 8B variants. Indices 1–12 correspond to the checkpoint ordering listed in Appendix A.

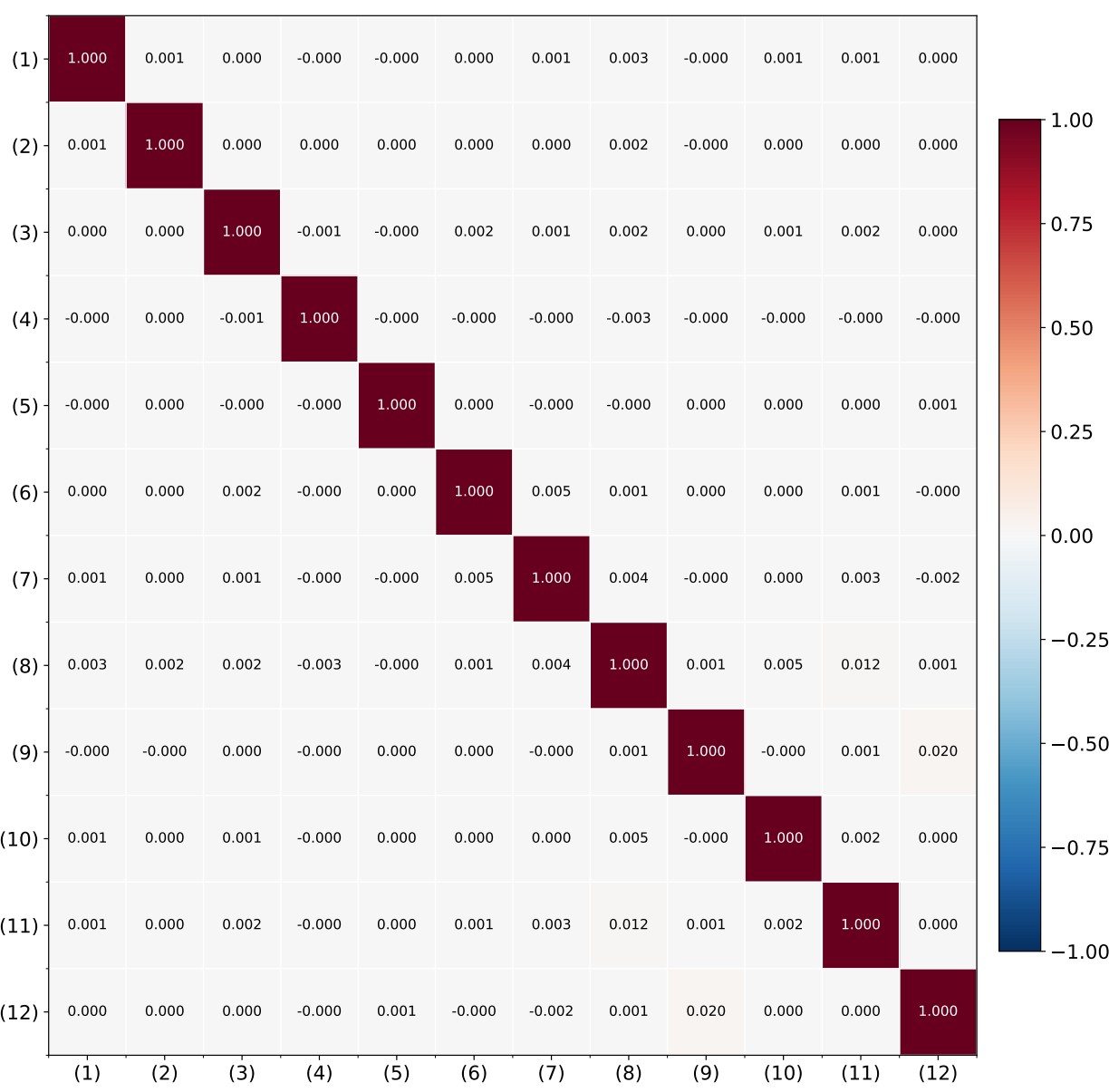

Figure 21: Pairwise cosine similarities between task vectors of fine-tuned QWEN3 4B variants. Indices 1–12 correspond to the checkpoint ordering listed in Appendix A.

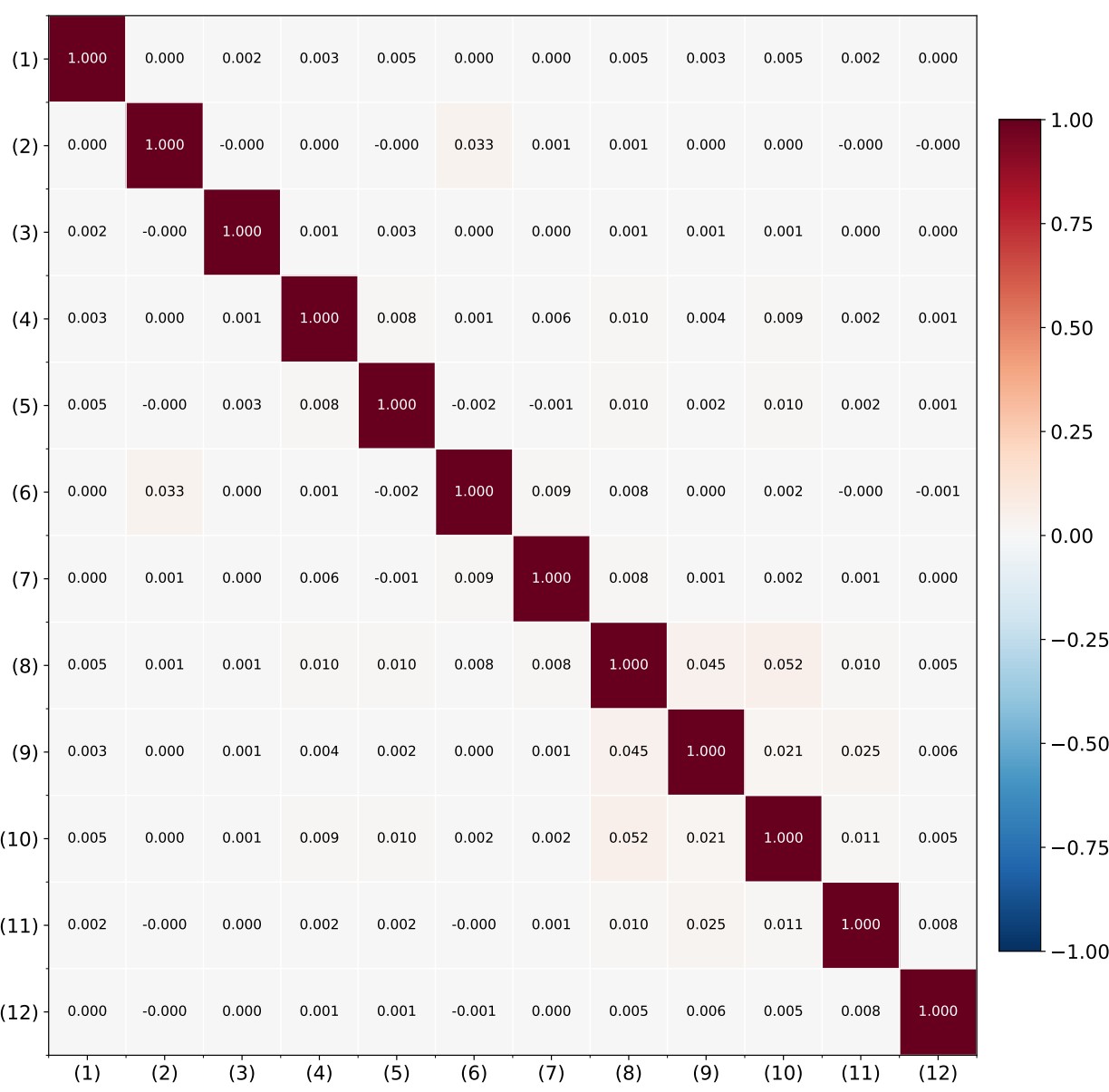

Figure 22: Pairwise cosine similarities between task vectors of fine-tuned QWEN3 8B variants. Indices 1–12 correspond to the checkpoint ordering listed in Appendix A.

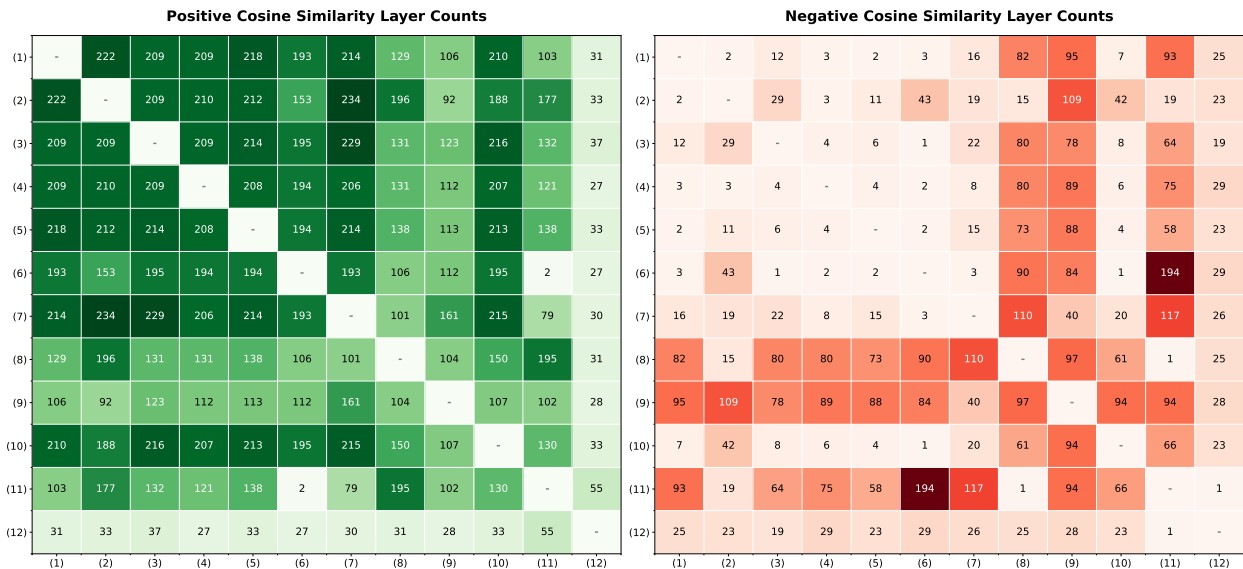

Figure 23: Layerwise sign heatmap of pairwise cosine similarities between task vectors of fine-tuned LLAMA 3.2 3B variants, showing the number of layers with positive and negative alignment. Indices 1–12 correspond to the checkpoint ordering in Appendix A.

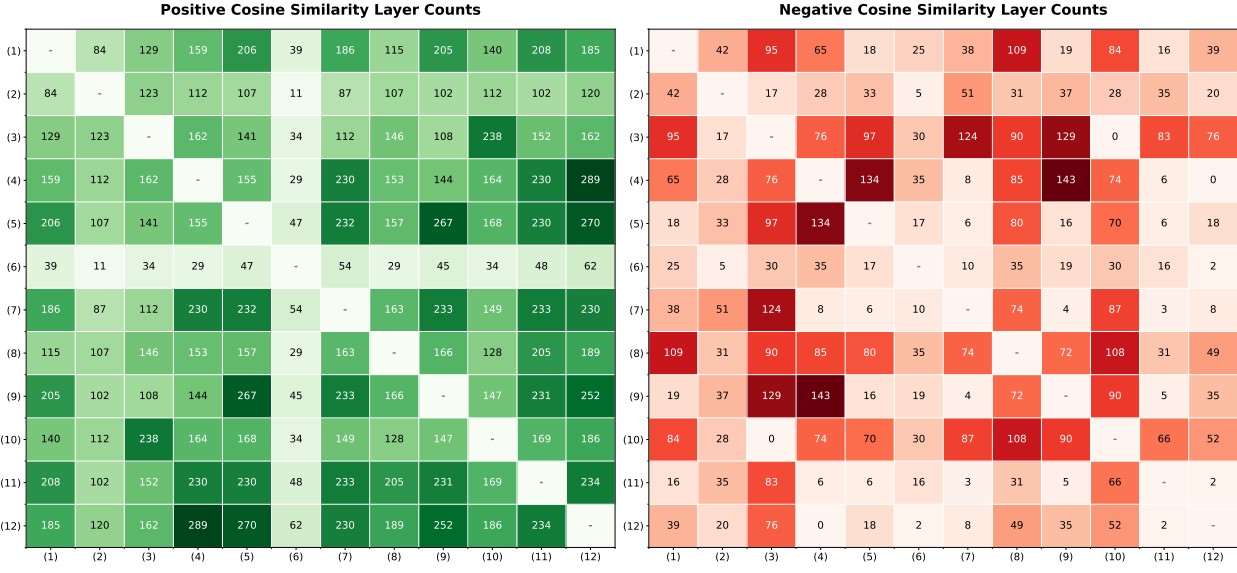

Figure 24: Layerwise sign heatmap of pairwise cosine similarities between task vectors of fine-tuned LLAMA 3.1 8B variants, showing the number of layers with positive and negative alignment. Indices 1–12 correspond to the checkpoint ordering in Appendix A.

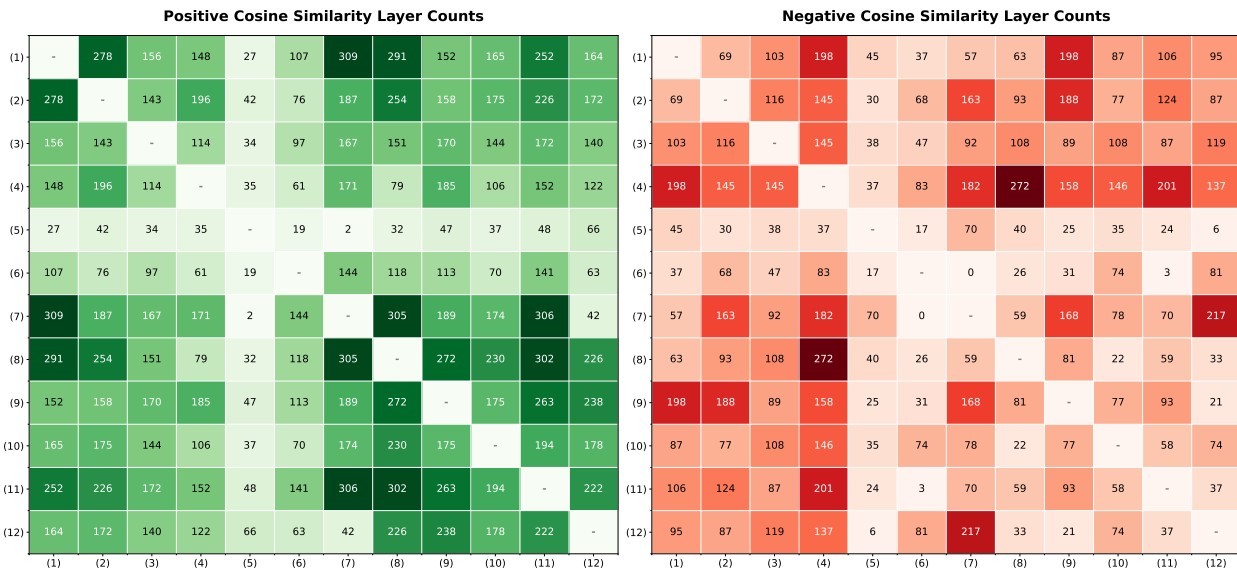

Figure 25: Layerwise sign heatmap of pairwise cosine similarities between task vectors of fine-tuned Qwen3 4B variants, showing the number of layers with positive and negative alignment. Indices 1–12 correspond to the checkpoint ordering in Appendix A.

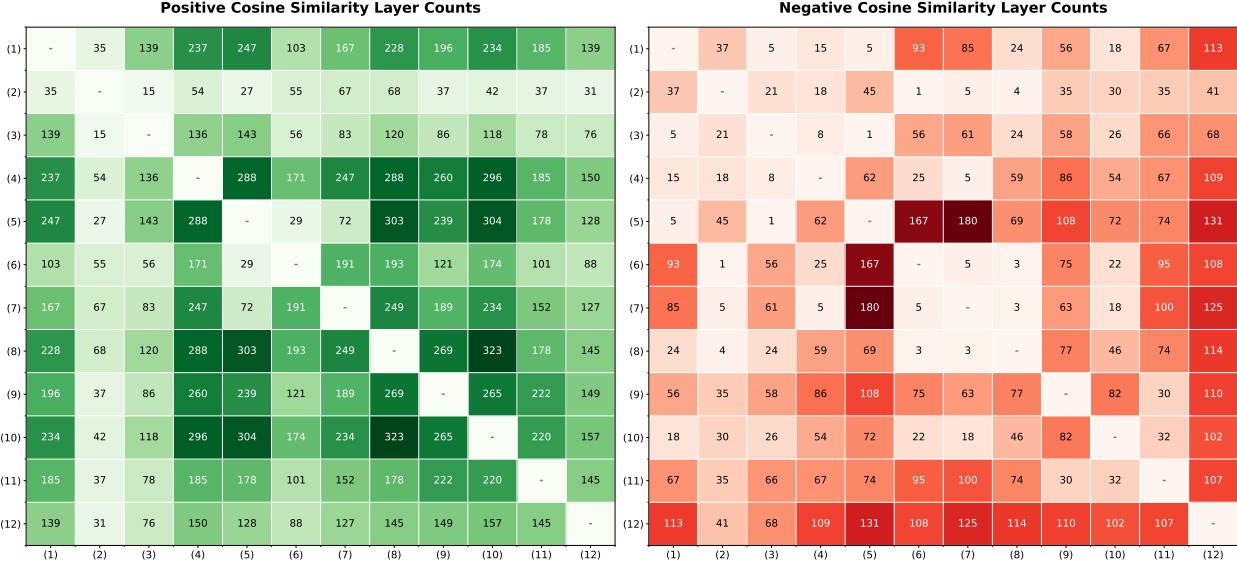

Figure 26: Layerwise sign heatmap of pairwise cosine similarities between task vectors of fine-tuned Qwen3 8B variants, showing the number of layers with positive and negative alignment. Indices 1–12 correspond to the checkpoint ordering in Appendix A.

# E   Experiments on Homogeneous Merges

In addition to the random subset sampling strategy presented in the main text, we investigate the behavior of merging methods when applied to domain-specific groups of experts. Specifically, we aim to determine whether merging models from distinct domains (Medical and Math) introduces destructive interference compared to merging experts from a single domain.

## E.1   Experimental Setup

For both QWEN3-4B and QWEN3-8B, we selected three fine-tuned checkpoints specialized in medical tasks and three specialized in mathematics from the Hugging Face Hub. The specific models used are listed in Table 4.

Table 4: List of domain-specific checkpoints used for homogeneous merging experiments.

| Base Model | Domain | Checkpoint Name |
|---|---|---|
| QWEN3 4B | Math | prithivMLmods/Draconis-Qwen3_Math-4B-Preview |
| | | AmberYifan/Qwen3-4B-MATH-GRPO-len-control |
| | | ertghiu256/qwen3-math-reasoner |
| | Medical | XformAI-india/Qwen3-4B-medicaldataset |
| | | mlxha/Qwen3-4B-grpo-medmcqa |
| | | Cannae-AI/MedicalQwen3-Reasoning-4B |
| QWEN3 8B | Math | taki555/Qwen3-8B-Shadow-FT-BAAI-2k |
| | | Jasaxion/MathSmith-Hard-Problem-Synthesizer-Qwen3-8B |
| | | mlfoundations-dev/a1_math_formulas |
| | Medical | mlxha/Qwen3-8B-grpo-medmcqa-v2 |
| | | BlueZeros/EHR-R1-8B |
| | | EricZhang1412/Qwen3-8B-SFT-MedicalExam |

We construct two domain-specific evaluation suites, Medical and Math, each composed of relevant subsets drawn from `mmlu`, `medmcqa`, `headqa`, and `bbh`. The specific task subsets included in each suite are listed in Table 5.

Table 5: Evaluation tasks used for homogeneous merging experiments, grouped by domain.

| Medical Tasks | Math Tasks |
|---|---|
| `medmcqa` | `mmlu_abstract_algebra` |
| `headqa_en` | `mmlu_college_mathematics` |
| `mmlu_anatomy` | `mmlu_elementary_mathematics` |
| `mmlu_clinical_knowledge` | `mmlu_formal_logic` |
| `mmlu_medical_genetics` | `leaderboard_bbh_boolean_expressions` |
| `mmlu_professional_medicine` | `leaderboard_bbh_formal_fallacies` |
| `mmlu_virology` | `leaderboard_bbh_geometric_shapes` |
| `mmlu_college_medicine` | `leaderboard_bbh_object_counting` |
| | `leaderboard_bbh_web_of_lies` |

To evaluate the impact of mixing domains, we performed three merge configurations for each base model and method: (i) **Merge Medical Only** ($n = 3$), which merges only the three medical experts and is evaluated on medical tasks; (ii) **Merge Math Only** ($n = 3$), which merges only the three math experts and is evaluated on math tasks; and (iii) **Merge All** ($n = 6$), which merges all six experts (three medical and three math) and is evaluated on both medical and math tasks.

### E.2 Results

The results for QWEN3 8B and QWEN3 4B are presented in Fig. 27 and Fig. 28. For QWEN3 8B, we observe stability across all merging methods. Specifically, models merged from all six experts perform nearly identically to those merged from domain-specific subsets. This indicates that, at this scale, combining disjoint domains like Math and Medicine introduces negligible interference, allowing the merged model to retain the specialized capabilities of both groups simultaneously.

A similar trend holds for QWEN3 4B. Standard approaches like Task Arithmetic and TIES effectively preserve task performance, yielding multi-domain models that match their single-domain counterparts. However, we observe more noticeable deviations in subspace-based variants. For instance, TA + SB and Iso-C exhibit larger performance gaps between single-domain and multi-domain merges, likely reflecting the increased sensitivity of subspace selection at smaller parameter scales. Despite these exceptions, the overall pattern reveals that diverse experts can generally be combined without significant negative transfer.

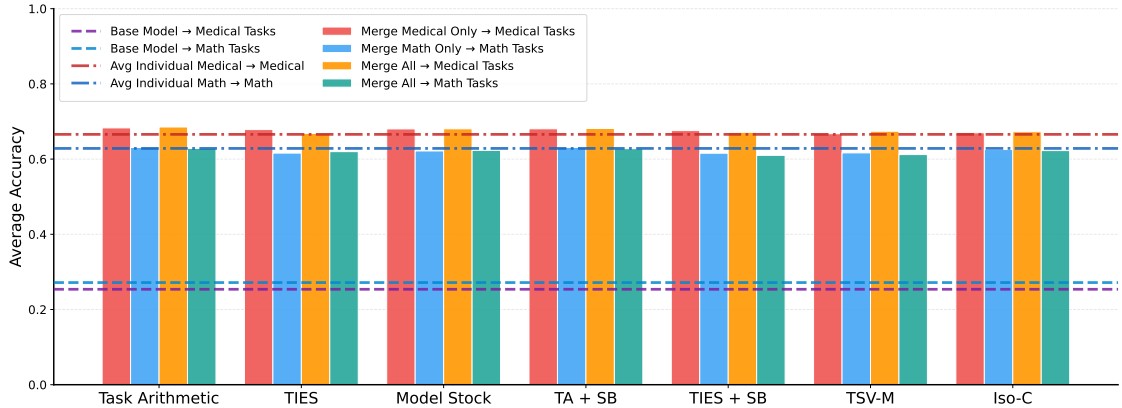

Figure 27: Performance comparison of homogeneous vs. heterogeneous merging on QWEN3-8B. We compare merging only domain-specific experts (Medical Only, Math Only) against merging all experts together.

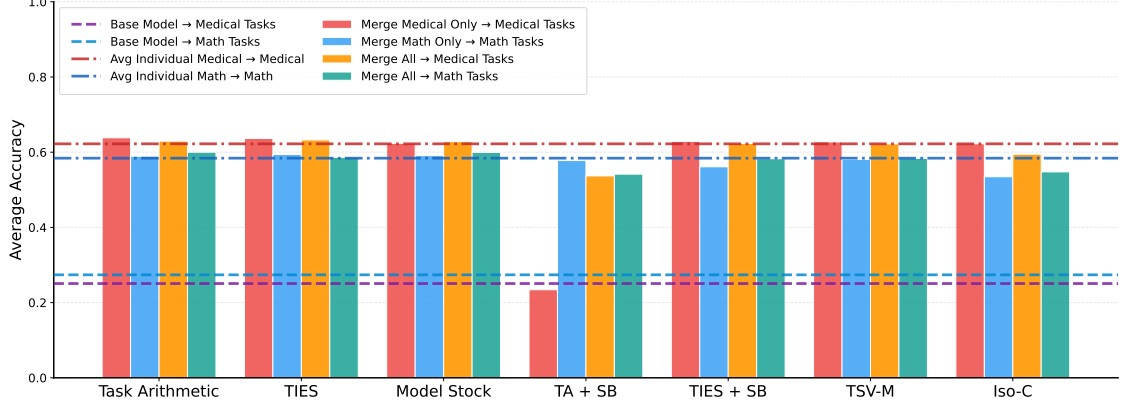

Figure 28: Performance comparison of homogeneous vs. heterogeneous merging on QWEN3-4B. We compare merging only domain-specific experts (Medical Only, Math Only) against merging all experts together.

## F  Evaluation Details and Configuration for *lm-eval-harness*

All evaluations follow the default *lm-eval-harness* inference and task configuration. Decoding is performed using the harness default generation setup. Models are evaluated using float16 precision with batch_size=auto. For each task, the number of in-context examples (n-fewshot) is determined by the task definition shipped with the harness. We do not apply model-specific chat templates, and no task-specific prompt engineering or template modifications are applied. Table 6 summarizes the n-fewshot values used for each benchmark.

| Task | n-fewshot |
|---|---|
| arc_easy | 0 |
| arc_challenge | 0 |
| boolq | 0 |
| commonsense_qa | 0 |
| hellaswag | 0 |
| winogrande | 0 |
| piqa | 0 |
| openbookqa | 0 |
| headqa | 0 |
| prost | 0 |
| truthfulqa_mc1 | 0 |
| medmcqa | 0 |
| leaderboard_gpqa | 0 |
| leaderboard_bbh | 3 |
| mmlu | 0 |
| leaderboard_mmlu_pro | 5 |

Table 6: Number of in-context examples (n-fewshot) used for each evaluation task, following the default configuration of lm-eval-harness.

