# OpenReview forum: "A Systematic Study of In-the-Wild Model Merging for Large Language Models"
_TMLR — Accepted by TMLR_

### Review · Reviewer_24CT · 2025-12-23

**Summary Of Contributions:**

This paper presents a systematic evaluation of model merging for LLMs, pairing each of four base models with 12 publicly available fine-tuned checkpoints, sampling multiple subset sizes, and evaluating merged models on standard benchmarks via LM-Eval-Harness. It compares six merging methods, three weight space and three subspace, with implementations applied through the mergekit library. The main empirical finding is that Task Arithmetic is the only method that consistently achieves constructive interference under the paper’s random-subset protocol. In contrast, interference-aware and subspace-based methods often degrade as more checkpoints are merged. The paper also reports a correlation between larger parameter space displacement from the base model and worse average benchmark accuracy, using the L2 norm of the merged update as a diagnostic.

Key strengths are the scale and consistency of the evaluation grid across model families and subset sizes. Key weaknesses include limited merging regimes and some reporting details needed for exact replication that are not fully specified in the main text.

**Audience:**

Yes

**Audience Explanation:**

The topic aligns with active interest in post-training reuse of open weight checkpoints, since the paper studies when merging can replace or reduce additional training across multiple LLM families and sizes. This paper provides a large, standardized comparison that clarifies which prior merging ideas transfer poorly to LLM settings under a realistic “public checkpoint soup” style protocol, which is directly relevant to researchers who use mergekit-style tooling.

**Broader Impact Concerns:**

This paper’s broader impact statement notes that merging can amplify undesirable behaviors, privacy risks, or misinformation when checkpoints have unknown provenance or biases, and it recommends caution and monitoring.

**Claims And Evidence:**

Yes

**Claims Explanation:**

* The core claim that Task Arithmetic improves reliably with more merged checkpoints, while TIES Merging often degrades, is directly supported by the average accuracy trends and the constructive interference tables across four base models.

 * The claim that merges can surpass not only the base but also the strongest individual fine-tuned checkpoint for three of four model families is directly supported by the reported comparisons against “Best FT.”

 * The negative results for subspace methods under the same protocol are also directly supported by the accuracy trends and the success probability and mean change table for subspace-based methods.

**Requested Changes:**

* The authors are suggested to add at least one additional merging regime beyond random subsets, such as similarity-based clustering of checkpoints or filtered subsets, and report whether the rankings of methods change under that regime.

* The authors are suggested to specify the evaluation harness configuration in sufficient detail to reproduce results, including few-shot settings, prompts or templates if applicable, and any decoding details for tasks.

* There is a typo in the caption of Table 1: “Arithmetic.”

---

### Review · Reviewer_62PW · 2025-12-27

**Summary Of Contributions:**

This submission conducts a broad empirical study of model merging for modern LLMs (e.g., Llama 3 and Qwen families) using a set of publicly available fine-tuned checkpoints and a wide benchmark suite. The paper’s central finding is that simple Task Arithmetic (TA) yields the most consistent performance improvements as the number of merged checkpoints increases, whereas more elaborate methods such as TIES and subspace-based approaches often degrade performance in the evaluated setting.

**Key strengths:** broad empirical coverage across model families/benchmarks; practical motivation aligned with "in-the-wild" checkpoint reuse; an attempt to provide a mechanistic interpretation via deviation-from-base diagnostics.

**Key weaknesses:** (i) baseline scope mismatch for Model Stock under highly heterogeneous merging, (ii) conclusions risk overgeneralization from the specific "random heterogeneous subset" regime, and (iii) potentially asymmetric hyperparameter tuning across methods, especially for TIES (top-k and scaling interaction).

**Additional Comments:**

**Reject**.
The paper is timely and potentially valuable, especially as a "reality check" for practitioners. At present, I believe the study’s empirical results are interesting but the broader interpretation is not yet fully supported due to (i) ambiguous baseline scope for Model Stock, (ii) insufficient control/measurement of checkpoint heterogeneity, and (iii) lack of distance-constrained and uniformly tuned comparisons. Addressing the critical items above would substantially strengthen the paper and make the conclusions more general and reliable.

**Audience:**

Yes

**Audience Explanation:**

The topic of model merging is highly relevant to the TMLR audience, given the increasing cost of training LLMs and the proliferation of fine-tuned checkpoints. A rigorous "reality check" on whether complex merging methods justify their complexity over simple averaging is valuable. If the experimental flaws are addressed, the finding that simple methods are robust would be of significant practical interest to engineers and researchers aiming to reuse public models efficiently.

**Broader Impact Concerns:**

The paper includes a Broader Impact Statement discussing the efficiency benefits and potential risks of uncontrolled merging. There are no additional specific ethical concerns requiring a new statement, provided the technical claims are corrected.

**Claims And Evidence:**

No

**Claims Explanation:**

The evidence provided does not sufficiently support the broad claim that modern merging methods "fail on LLMs" due to three critical flaws in the experimental design:
1. **Heterogeneity vs. Method Failure**: The evidence convincingly supports a *narrower* claim: *under heterogeneous, randomly mixed checkpoint merging, TA appears more robust than TIES/subspace methods in the authors’ evaluation pipeline.* However, the submission at times reads as making a broader claim (e.g., that these methods "fail for LLMs" generally), while the experiments do not sufficiently disentangle LLM-specific failure from strong heterogeneity-driven failure.
2. **Misapplication of Baselines (Model Stock)**: One baseline (Model Stock) is evaluated primarily in a regime that does not match its intended scope (merging models trained under the same task/setup with different random seeds). In such an out-of-scope regime, underperformance is difficult to interpret as evidence against the method, which weakens the clarity and fairness of the comparative conclusions.
3. **Unfair Hyperparameter Tuning**: The analysis attributes poor performance (especially for TIES) to large deviation from the base model, yet the tuning protocol does not clearly ensure that TIES method is given comparable opportunity to control distance via scaling (λ), unlike TA. Without distance-constrained or distance-matched controls (and/or a fair tuning budget across methods), the causal explanation remains under-validated.

**Requested Changes:**

To secure a recommendation for acceptance, the following critical adjustments are required:
1. **Rectify the Model Stock Comparison (Critical)**: The evaluation of Model Stock on heterogeneous tasks must be fundamentally revised. Either remove Model Stock from the main comparative analysis or explicitly frame it as an out-of-distribution stress test, acknowledging that the setup violates the method's intended scope (same-task merging). The current framing implies a failure of the method itself, which is misleading.
2. **Control for Heterogeneity (Critical)**: To support the claim that these methods fail on LLMs (rather than just on conflicting tasks), the authors must add experiments with homogeneous merging.
    - Cluster checkpoints by domain (e.g., merge only Math models, or only Coding models) and evaluate TIES/Subspace methods in this setting.
    - If TIES/Subspace methods work well in homogeneous settings but fail in random settings, the paper's conclusion must change to reflect that these methods lack robustness to task conflict, not that they are incompatible with LLMs.
3. **Implement Fair Tuning Protocols (Critical)**: The comparison between TA and TIES/Subspace methods must be standardized regarding hyperparameter optimization.
    - Perform a grid search for TIES over both density (k) and scaling factor (λ).
    - Specifically, conduct a "Distance-Constrained" comparison: Tune the scaling factor of TIES/Subspace methods so that their final merged model has a similar L2 distance from the base model as the TA model. Compare performance at this iso-distance point to determine if the failure is due to the direction of the update or simply the magnitude.
4. **Refine Claims**: Soften the broad claims about method failure to be specific to the experimental context (random heterogeneous mixing) until the above controls are implemented.

---

### Review · Reviewer_AyLX · 2026-01-08

**Summary Of Contributions:**

The paper is an empirical study evaluating recent model merging methods on various LLMs.

Strengths:
1. The paper provides a much needed systematic evaluation of recent model merging methods on recent open-weights LLMs.

Weaknesses:
1. In my opinion, the evaluation protocol presented in the paper entangles two independent factors: (a) merging of LLMs, and (b) merging "in the wild". The authors claim to investigate (a) merging of LLMs, but (b) merging "in the wild" may obscure it.
   Most merging methods implicitly assume that each checkpoint is useful and can positively contribute to the final merged model. It is always the case in the standard controlled evaluation protocols of multi-task image classification or natural language inference. However, when selecting some checkpoints from an open model hub, it is not clear if the model can positively contribute. Therefore, the superiority of one merging method over the other, can come from the fact that it secretly reduces the influence of "harming checkpoints".
2. Main results are questionable.
	1. In Fig. 3 and Fig. 6, "Baseline" (which I assume is an evaluation of a base model) is significantly lower than merging 0 models. They should be exactly equal which indicates some inconsistency in the evaluation protocol.
	2. Figure 4: How is the L2-norm of Task Arithmetic decreasing with the number of merged models? For a constant $\lambda=1$, it could happen only if the task vectors had significant opposite components that would manifest itself in negative cosine similarity between task vectors. For cos_sim >= 0, L2-norm grows between $\sqrt n$ (cos_sim = 0) and $n$ (cos_sim = 1), where $n$ is the number of merged task vectors.
3. Hyperparameters selection:
	1. I am very sceptical of Figure 11. It contradicts previous results (eg TIES merging paper, Fig 8 left), my own experiments for merging LLMs and common sense: summing 12 task vectors and scaling the sum by 1.9 achieves the best results for llama models. This does not make much sense.
	2. What values of $\lambda$ are used for other merging methods?
4. Phrases like "multi-talented model" seem LLM-generated
5. Model Stock is a method designed to merge models fine-tuned on the same dataset with the same hyperparameters. It is an odd baseline for multi-task merging "in the wild".

**Audience:**

Yes

**Audience Explanation:**

I believe that the results could be interesting for model merging community **once the evaluation issues are resolved**. In the current state, I find the results presented in the paper confusing.

**Broader Impact Concerns:**

In my opinion, Broader Impact Statement provided by the paper is sufficient.

**Claims And Evidence:**

No

**Claims Explanation:**

In my opinion, the current evaluation setup entangles two orthogonal problems: merging of LLMs and merging "in the wild" (see Weakness 1). Therefore, the claims about some merging methods working or not working for LLMs are not necessarily true. Moreover, some of the presented results are not convincing (see Weakness 2 and 3).

**Requested Changes:**

Critical changes:
1. Make evaluation protocol more clear (see Weakness 1). I can see 2 possible ways of achieving it. First (better) would be to train the models on tasks related to the evaluation benchmarks. Second would be to evaluate the individual models and filtered the ones that do not improve over the base model. Once we know which merging methods work for LLMs,  the current setup could serve as an extension -- it would tell which methods are robust to inclusion of harmful checkpoints.
2. Address evaluation issues specified in Weakness 2
3. Clarify hyperparameters selection (Weakness 3).

Improvements:
1. Among the most intriguing observations is the phenomenon of constructive interference, where a merged model outperforms its individual base models. -- this statement would benefit from citation
2. Section 2.2 -- TIES-Merging is missing citation
3. Fig 2.2 -- first step in TIES-Merging diagram is captioned as "Task Vectors Layerwise" while it should be "Task Vectors Parameter-wise"
4. Table 2 -- it is unclear what "best fine-tuned checkpoint" is. Is it a single checkpoint that performed best on average across all the evaluated tasks or is it the best model for each evaluation task?

---

### Decision · Action_Editor_75yT · 2026-03-11

**Recommendation:** Accept with minor revision

**Additional Comments:**

I found a few minor issues that need to be corrected in the final version of the paper.

1. There appears to be a typo in the following sentence from the introduction, since the construction "work on LLMs, also improve performance of “in-the-wild” merging in LLMs?" is quite awkward. "(3) Do recently proposed merging methods that operate on the subspaces of weight matrices work on LLMs, also improve performance of “in-the-wild” merging in LLMs?"

2. There are several minor errors in the bibliography:

- Charles Goddard, Shamane Siriwardhana, Malikeh Ehghaghi, Luke Meyers, Vlad Karpukhin, Brian Benedict, Mark McQuade, and Jacob Solawetz. Arcee’s mergekit: A toolkit for merging large language models. -- appeared at EMNLP 2024 in the industry track
- Pavel Izmailov, Dmitrii Podoprikhin, Timur Garipov, Dmitry Vetrov, and Andrew Gordon Wilson. Averaging weights leads to wider optima and better generalization -- appeared at UAI 2018
- W. Li, Y. Peng, M. Zhang, L. Ding, H. Hu and L. Shen, "Deep Model Fusion: A Survey," in IEEE Transactions on Neural Networks and Learning Systems, doi: 10.1109/TNNLS.2025.3628666.
- The full reference for Shoemake1985 is

@inproceedings{shoemake1985animating,
  title={Animating rotation with quaternion curves},
  author={Shoemake, Ken},
  booktitle={Proceedings of the 12th annual conference on Computer graphics and interactive techniques},
  pages={245--254},
  year={1985}
}

- Prateek Yadav, Tu Vu, Jonathan Lai, Alexandra Chronopoulou, Manaal Faruqui, Mohit Bansal, and Tsendsuren Munkhdalai. What matters for model merging at scale - Appeared in TMLR (https://openreview.net/forum?id=9sbetmvNpW)

- The full reference for Yang2026 is

@article{yang2026model,
  title={Model merging in llms, mllms, and beyond: Methods, theories, applications, and opportunities},
  author={Yang, Enneng and Shen, Li and Guo, Guibing and Wang, Xingwei and Cao, Xiaochun and Zhang, Jie and Tao, Dacheng},
  journal={ACM Computing Surveys},
  volume={58},
  number={8},
  pages={1--41},
  year={2026},
  publisher={ACM New York, NY}
}

**Audience:**

Yes

**Audience Explanation:**

The scale of the experiments, range of merging algorithms tested, and use of mergekit in the evaluations will be of interest to other researchers.

**Claims And Evidence:**

Yes

**Claims Explanation:**

The primary claim of the paper is that, out of six model merging algorithms examined, only Task Arithmetic produces merged LLMs that outperform the checkpoints used in the merging process in the "in-the-wild" setting where the LLM checkpoints are the result of fine tuning on heterogeneous and possibly overlapping tasks.

The paper provides experimental results for four different LLMs, spanning two model families and three model sizes, with evaluation on 16 standard benchmarks.

Properly framing the paper contribution was a major concern in the initial reviews, as the first revision of the paper did not clearly state that the setting was in-the-wild merging.